# Constrained Synthesis with Projected Diffusion Models

**Jacob K. Christopher**
University of Virginia
csk4sr@virginia.edu

**Stephen Baek**
University of Virginia
baek@virginia.edu

**Ferdinando Fioretto**
University of Virginia
fioretto@virginia.edu

## Abstract

This paper introduces an approach to endow generative diffusion processes the ability to satisfy and certify compliance with constraints and physical principles. The proposed method recast the traditional sampling process of generative diffusion models as a constrained optimization problem, steering the generated data distribution to remain within a specified region to ensure adherence to the given constraints. These capabilities are validated on applications featuring both convex and challenging, non-convex, constraints as well as ordinary differential equations, in domains spanning from synthesizing new materials with precise morphometric properties, generating physics-informed motion, optimizing paths in planning scenarios, and human motion synthesis.

## 1 Introduction

Generative diffusion models excel at robustly synthesizing content from raw noise through a sequential denoising process [14, 24]. They have revolutionized high-fidelity creation of complex data, and their applications have rapidly expanded beyond mere image synthesis, finding relevance in areas such as engineering [29, 34], automation [3, 16], chemistry [1, 15], and medical analysis [2, 6]. However, although diffusion models excel at generating content that is coherent and aligns closely with the original data distribution, their direct application in scenarios requiring stringent adherence to predefined criteria poses significant challenges. Particularly the use of diffusion models in scientific and engineering domains where the generated data needs to not only resemble real-world examples but also rigorously comply with established specifications and physical laws remains an open challenge.

Given these limitations, one might consider training a diffusion model on a dataset that already adheres to specific constraints. However, even with "feasible" training data, this approach does not guarantee adherence to desired criteria due to the stochastic nature of the diffusion process. Furthermore, there are frequent scenarios where the training data must be altered to generate outputs that align with specific properties, potentially not present in the original data. This issue often leads to a distribution shift further exacerbating the inability of generative models to produce "valid" data. As we will show in a real-world experiment (§6.1), this challenge is particularly acute in scientific and engineering domains, where training data is often sparse and confined to specific distributions, yet the synthesized outputs are required to meet stringent properties or precise standards [29].

This paper addresses these challenges and introduces *Projected Diffusion Models* (PDM), a novel approach that recast the traditional sampling strategy in diffusion processes as a constrained-optimization problem. This perspective allows us to apply traditional techniques from constraint optimization to the sampling process. In this work, the problem is solved by iteratively projecting the diffusion sampling process onto arbitrary constraint sets, ensuring that the generated data adheres strictly to imposed constraints or physical principles. We provide theoretical support for PDM's capability to not only certify adherence to the constraints but also to optimize the generative model's original objective of replicating the true data distribution. This alignment is a significant advantage of PDM, yielding state-of-the-art FID scores while maintaining strict compliance with the imposed constraints.

38th Conference on Neural Information Processing Systems (NeurIPS 2024).

**Contributions.** In summary, this paper makes the following key contributions: **(1)** It introduces PDM, a new framework that augments diffusion-based synthesis with arbitrary constraints in order to generate content with high fidelity that also adheres to the imposed specifications. The paper elucidates the theoretical foundation that connects the reverse diffusion process to an optimization problem, facilitating the direct incorporation of constraints into the reverse process of score-based diffusion models. **(2)** Extensive experiments across various domains demonstrate PDM's effectiveness. These include adherence to morphometric properties in real-world material science experiments, physics-informed motion governed by ordinary differential equations, trajectory optimization in motion planning, and constrained human motion synthesis, showcasing PDM's ability to produce content that adheres to both complex constraints and physical principles. **(3)** We further show that PDM is able to generate out-of-distribution samples that meet stringent constraints, even in scenarios with extremely sparse training data and when the training data does not satisfy the required constraints. **(4)** Finally, we provide a theoretical basis elucidating the ability of PDM to generate highly accurate content while ensuring constraint compliance, underpinning the practical implications of this approach.

## 2 Preliminaries: Diffusion models

Diffusion-based generative models [14, 24] expand a data distribution, whose samples are denoted $\boldsymbol{x}_0$, through a Markov chain parameterization $\{\boldsymbol{x}_t\}_{t=1}^T$, defining a Gaussian diffusion process $p(\boldsymbol{x}_0) = \int p(\boldsymbol{x}_T) \prod_{t=1}^T p(\boldsymbol{x}_{t-1}|\boldsymbol{x}_t)d\boldsymbol{x}_{1:T}$.

In the *forward process*, the data is incrementally perturbed towards a Gaussian distribution. This process is represented by the transition kernel $q(\boldsymbol{x}_t|\boldsymbol{x}_{t-1}) = \mathcal{N}(\boldsymbol{x}_t; \sqrt{1-\beta_t}\boldsymbol{x}_{t-1}, \beta_t\boldsymbol{I})$ for some $0 < \beta_t < 1$, where the $\beta$-schedule $\{\beta_t\}_{t=1}^T$ is chosen so that the final distribution $p(\boldsymbol{x}_T)$ is nearly Gaussian. The diffusion time $t$ allows an analytical expression for variable $\boldsymbol{x}_t$ represented by $\chi_t(\boldsymbol{x}_0, \epsilon) = \sqrt{\alpha_t}\boldsymbol{x}_0 + \sqrt{1-\alpha_t}\epsilon$, where $\epsilon \sim \mathcal{N}(\boldsymbol{0}, \boldsymbol{I})$ is a noise term, and $\alpha_t = \prod_{i=1}^t (1-\beta_i)$. This process is used to train a neural network $\epsilon_\theta(\boldsymbol{x}_t, t)$, called the *denoiser*, which implicitly approximates the underlying data distribution by learning to remove noise added throughout the forward process. The training objective minimizes the error between the actual noise $\epsilon$ and the predicted noise $\epsilon_\theta(\chi_t(\boldsymbol{x}_0, \epsilon), t)$ via the loss function:

$$\min_\theta \mathbb{E}_{t\sim[1,T],\ p(\boldsymbol{x}_0),\mathcal{N}(\epsilon;\boldsymbol{0},\boldsymbol{I})} \left[ \|\epsilon - \epsilon_\theta(\chi_t(\boldsymbol{x}_0,\epsilon),t)\|^2 \right]. \tag{1}$$

The *reverse process* uses the trained denoiser, $\epsilon_\theta(\boldsymbol{x}_t, t)$, to convert random noise $p(\boldsymbol{x}_T)$ iteratively into realistic data from distribution $p(\boldsymbol{x}_0)$. Practically, $\epsilon_\theta$ predicts a single step in the denoising process that can be used during sampling to reverse the diffusion process by approximating the transition $p(\boldsymbol{x}_{t-1}|\boldsymbol{x}_t)$ at each step $t$.

**Score-based models** [25, 26], while also operating on the principle of gradually adding and removing noise, focus on directly modeling the gradient (score) of the log probability of the data distribution at various noise levels. The score function $\nabla_{\boldsymbol{x}_t} \log p(\boldsymbol{x}_t)$ identifies the direction and magnitude of the greatest increase in data density at each noise level. The training aims to optimize a neural network $\mathbf{s}_\theta(\boldsymbol{x}_t, t)$ to approximate this score function, minimizing the difference between the estimated and true scores of the perturbed data:

$$\min_\theta \mathbb{E}_{t\sim[1,T],p(\boldsymbol{x}_0),q(\boldsymbol{x}_t|\boldsymbol{x}_0)}(1-\alpha_t) \left[ \|\mathbf{s}_\theta(\boldsymbol{x}_t,t) - \nabla_{\boldsymbol{x}_t} \log q(\boldsymbol{x}_t|\boldsymbol{x}_0)\|^2 \right], \tag{2}$$

where $q(\boldsymbol{x}_t|\boldsymbol{x}_0) = \mathcal{N}(\boldsymbol{x}_t; \sqrt{\alpha_t}\boldsymbol{x}_0, (1-\alpha_t)\boldsymbol{I})$ defines a distribution of perturbed data $\boldsymbol{x}_t$, generated from the training data, which becomes increasingly noisy as $t$ approach $T$. This paper considers score-based models.

## 3 Related work and limitations

While diffusion models are highly effective in producing content that closely mirrors the original data distribution, the stochastic nature of their outputs act as an impediment when specifications or constraints need to be imposed on the generated outputs. In an attempt to address this issue, two main approaches could be adopted: (1) model conditioning and (2) post-processing corrections.

**Model conditioning** [13] aims to control generation by augmenting the diffusion process via a conditioning variable $\boldsymbol{c}$ to transform the denoising process via classifier-free guidance:

$$\hat{\epsilon}_\theta \overset{\text{def}}{=} \lambda \times \epsilon_\theta(\boldsymbol{x}_t, t, \boldsymbol{c}) + (1 - \lambda) \times \epsilon_\theta(\boldsymbol{x}_t, t, \perp),$$

where $\lambda \in (0, 1)$ is the *guidance scale* and $\perp$ is a null vector representing non-conditioning. These methods have been shown effective in capturing properties of physical design [29], positional awareness [3], and motion dynamics [32]. However, while conditioning may be effective to influence the generation process, it lacks the rigor to ensure adherence to specific constraints. This results in generated outputs that, despite being plausible, may not be accurate or reliable. Figure 1 (red colors) illustrates this issue on a physics-informed motion experiment (detailed in §6.4). The figure reports the distance of the model outputs to feasible solutions, showcasing the constraint violations identified in a conditional model's outputs. Notably, the model, conditioned on labels corresponding to positional constraints, fails to generate outputs that adhere to these constraints, resulting in outputs that lack meaningful physical interpretation.

Additionally, conditioning in diffusion models often requires training supplementary classification and regression models, a process fraught with its own set of challenges. This approach demands the acquisition of extra labeled data, which can be impractical or unfeasible in specific scenarios. For instance, our experimental analysis will demonstrate a situation in material science discovery where the target property is well-defined, but the original data distribution fails to embody this property. This scenario is common in scientific applications, where data may not naturally align with desired outcomes or properties [19].

Figure 1: Sampling steps failing to converge to feasible solutions in conditional models (red) while minimizing the constraint divergence to 0 under PDM (blue).

**Post-processing correction.** An alternative approach involves applying post-processing steps to correct deviations from desired constraints in the generated samples. This correction is typically implemented in the last noise removal stage, $s_\theta(\boldsymbol{x}_1, 1)$. Some approaches have augmented this process to use optimization solvers to impose constraints on synthesized samples [10, 19, 21]. However these approaches present two main limitations. First, their objective does not align with optimizing the score function. This inherently positions the diffusion model's role as ancillary, with the final synthesized data often resulting in a significant divergence from the learned (and original) data distributions, as we will demonstrate in §6. Second, these methods are reliant on a limited and problem specific class of objectives and constraints, such as specific trajectory "constraints" or shortest path objectives which can be integrated as a post-processing step [10, 21].

**Other methods.** Some methods explored modifying either diffusion training or inference to adhere to desired properties. For instance, the methods in [9] and [17], support simple linear or convex sets, respectively. Similarly, Fishman et al. [7, 8] focus on predictive tasks within convex polytope, which are however confined to approximations by simple geometries like L2-balls. While important contributions, these approaches prove insufficient for the complex constraints present in many real-world tasks. Conversely, in the domain of image sampling, Lou and Ermon [18] and Saharia et al. [23] introduce methods like reflections and clipping to control numerical errors and maintain pixel values within the standard [0,255] range during the reverse diffusion process. These techniques, while enhancing sampling accuracy, do not address broader constraint satisfaction challenges.

To overcome these gaps and handle arbitrary constraints, our approach casts the reverse diffusion process to a constraint optimization problem that is then solved throught repeated projection steps.

## 4 Constrained generative diffusion

This section establishes a theoretical framework that connects the reverse diffusion process as an optimization problem. This perspective facilitates the incorporation of constraints directly into the process, resulting in the constrained optimization formulation presented in Equation (6).

The application of the reverse diffusion process of score-based models is characterized by iteratively transforming the initial noisy samples $\boldsymbol{x}_T$ back to a data sample $\boldsymbol{x}_0$ following the learned data distribution $q(\boldsymbol{x}_0)$. This transformation is achieved by iteratively updating the sample using the

estimated score function $\nabla_{\boldsymbol{x}_t} \log q(\boldsymbol{x}_t | \boldsymbol{x}_0)$, where $q(\boldsymbol{x}_t | \boldsymbol{x}_0)$ is the data distribution at time $t$. At each time step $t$, starting from $\boldsymbol{x}_t^0$, the process performs $M$ iterations of *Stochastic Gradient Langevin Dynamics* (SGLD) [30]:

$$\boldsymbol{x}_t^{i+1} = \boldsymbol{x}_t^i + \gamma_t \nabla_{\boldsymbol{x}_t^i} \log q(\boldsymbol{x}_t^i | \boldsymbol{x}_0) + \sqrt{2\gamma_t}\boldsymbol{\epsilon}, \tag{3}$$

where $\boldsymbol{\epsilon}$ is standard normal, $\gamma_t > 0$ is the step size, and $\nabla_{\boldsymbol{x}_t^i} \log q(\boldsymbol{x}_t^i | \boldsymbol{x}_0)$ is approximated by the learned score function $\mathbf{s}_\theta(\mathbf{x}_t, t)$.

## 4.1 Casting the reverse process as an optimization problem

First note that SGLD is derived from discretizing the continuous-time Langevin dynamics, which are governed by the stochastic differential equation:

$$d\boldsymbol{X}(t) = \nabla \log q(\boldsymbol{X}(t)) \, dt + \sqrt{2} \, d\boldsymbol{B}(t), \tag{4}$$

where $\boldsymbol{B}(t)$ is standard Brownian motion. Under appropriate conditions, the stationary distribution of this process is $q(\boldsymbol{x}_t)$ [22], implying that samples generated by Langevin dynamics will, over time, be distributed according to $q(\boldsymbol{x}_t)$. In practice, these dynamics are simulated using a discrete-time approximation, leading to the SGLD update in Equation (3). Therein the noise term $\sqrt{2\gamma_t}\,\boldsymbol{\epsilon}_t^i$ allows the algorithm to explore the probability landscape and avoid becoming trapped in local maxima.

Next notice that, as detailed in [30, 31], under some regularity conditions this iterative SGLD algorithm converges toward a stationary point, bounded by $\frac{d^2}{\sigma^{1/4}\lambda^*} \log(1/\epsilon)$, where, $\sigma^2$ represents the variance schedule, $\lambda^*$ denotes the uniform spectral gap of the Langevin diffusion, and $d$ is the dimensionality of the problem. Thus, as the reverse diffusion process progresses towards $T \to 0$, and the variance schedule decreases, the stochastic component becomes negligible, and SGLD transitions toward deterministic gradient ascent on $\log q(\boldsymbol{x}_t)$. In the limit of vanishing noise, the update rule simplifies to:

$$\boldsymbol{x}_t^{i+1} = \boldsymbol{x}_t^i + \gamma_t \nabla_{\boldsymbol{x}} \log q(\boldsymbol{x}_t^i | \boldsymbol{x}_0), \tag{5}$$

which is standard gradient ascent aiming to maximize $\log q(\boldsymbol{x}_t)$. This allow us to view the reverse diffusion process as an optimization problem minimizing the negative log-likelihood of the data distribution $q(\boldsymbol{x}_t | \boldsymbol{x}_0)$ at each time step $t$.

In traditional score-based models, at any point throughout the reverse process, $\boldsymbol{x}_t$ is *unconstrained*. When these samples are required to satisfy some constraints, the objective remains unchanged, but the solution to this optimization must fall within a feasible region $\mathbf{C}$, and thus the optimization problem formulation becomes:

$$\underset{\boldsymbol{x}_T, \ldots, \boldsymbol{x}_1}{\text{minimize}} \sum_{t=T,\ldots,1} -\log q(\boldsymbol{x}_t | \boldsymbol{x}_0) \tag{6a}$$

$$\text{s.t.:} \quad \boldsymbol{x}_T, \ldots, \boldsymbol{x}_0 \in \mathbf{C}. \tag{6b}$$

Operationally, the negative log likelihood is minimized at each step of the reverse Markov chain, as the process transitions from $\boldsymbol{x}_T$ to $\boldsymbol{x}_0$. In this regard, and importantly, the objective of the PDM's reverse sampling process is aligned with that of traditional score-based diffusion models.

## 4.2 Constrained guidance through iterative projections

The score network $\mathbf{s}_\theta(\boldsymbol{x}_t, t)$ directly estimates the first-order derivatives of Equation (6a), providing the necessary gradients for iterative gradient-based updates defined in Equation (3). In the presence of constraints (6b), however, an alternative iterative method is necessary to guarantee feasibility. PDM models a projected guidance approach to provide this constraint-aware optimization process.

First, we define the projection operator, $\mathcal{P}_{\mathbf{C}}$, as a constrained optimization problem,

$$\mathcal{P}_{\mathbf{C}}(\boldsymbol{x}) = \underset{\boldsymbol{y} \in \mathbf{C}}{\operatorname{argmin}} \, ||\boldsymbol{y} - \boldsymbol{x}||_2^2, \tag{7}$$

that finds the nearest feasible point to the input $\boldsymbol{x}$. The *cost of the projection* $||\boldsymbol{y} - \boldsymbol{x}||_2^2$ represents the distance between the closest feasible point and the original input.

To retain feasibility through an application of the projection operator after each update step, the paper defines *projected diffusion model sampling* step as

$$\boldsymbol{x}_t^{i+1} = \mathcal{P}_{\mathbf{C}} \left( \boldsymbol{x}_t^i + \gamma_t \nabla_{\boldsymbol{x}_t^i} \log q(\boldsymbol{x}_t|\boldsymbol{x}_0) + \sqrt{2\gamma_t}\boldsymbol{\epsilon} \right), \tag{8}$$

where $\mathbf{C}$ is the set of constraints and $\mathcal{P}_{\mathbf{C}}$ is a projection onto $\mathbf{C}$. Hence, iteratively throughout the Markov chain, a gradient step is taken to minimize the objective defined by Equation (6a) while ensuring feasibility. Convergence is guaranteed for convex constraints sets [20] and empirical evidence in §6 showcases the applicability of this methods to arbitrary constraint sets. Importantly, the projection operators can be *warm-started* during the repeated sampling step providing a piratical solution even for hard non-convex constrained regions. The full sampling process is detailed in Algorithm 1.

---

**Algorithm 1:** PDM

1   $\boldsymbol{x}_T^0 \sim \mathcal{N}(\mathbf{0}, \sigma_T \boldsymbol{I})$
2   **for** $t = T$ **to** $1$ **do**
3      $\gamma_t \leftarrow \sigma_t^2/2\sigma_T^2$
4      **for** $i = 1$ **to** $M$ **do**
5          $\boldsymbol{\epsilon} \sim \mathcal{N}(\mathbf{0}, \boldsymbol{I})$;
          $\boldsymbol{g} \leftarrow \boldsymbol{s}_{\theta^*}(\boldsymbol{x}_t^{i-1}, t)$
6          $\boldsymbol{x}_t^i = \mathcal{P}_{\boldsymbol{C}}(\boldsymbol{x}_t^{i-1} + \gamma_t \boldsymbol{g} + \sqrt{2\gamma_t}\boldsymbol{\epsilon})$
7      $\boldsymbol{x}_{t-1}^0 \leftarrow \boldsymbol{x}_t^M$
8   **return** $\boldsymbol{x}_0^0$

---

By incorporating constraints throughout the sampling process, the interim learned distributions are steered to comply with these specifications. This is empirically evident from the pattern in Figure 1 (blue curves): remarkably, the constraint violations decrease with each addition of estimated gradients and noise and approaches 0-violation as $t$ nears zero. *This trend not only minimizes the impact but also reduces the optimality cost of projections applied in the later stages of the reverse process.* We provide theoretical rationale for the effectiveness of this approach in §5 and conclude this section by noting that this approach can be clearly distinguished from other methods which use a diffusion model's sampling process to generate starting points for a constrained optimization algorithm [10, 21]. Instead, PDM leverages minimization of negative log likelihood as the primary objective of the sampling algorithm akin to standard unconstrained sampling procedures. This strategy offers a key advantage: *the probability of generating a sample that conforms to the data distribution is optimized directly*, rather than an external objective, *while simultaneously imposing verifiable constraints*. In contrast, existing baselines often neglect the conformity to the data distribution, which, as we will show in the next section, can lead to a deviation from the learned distribution and an overemphasis on external objectives for solution generation, resulting in significant divergence from the data distribution, reflected by high FID scores.

## 5   Effectiveness of PDM: A theoretical justification

Next, we theoretically justify the use of iterative projections to guide the sample to the constrained distribution. The analysis assumes that the feasible region $C$ is a convex set. All proofs are reported in the Appendix. We start by defining the update step.

**Definition 5.1.** The operator $\mathcal{U}$ defines a single update step for the sampling process as,

$$\mathcal{U}(\boldsymbol{x}_t^i) = \boldsymbol{x}_t^i + \gamma_t \mathbf{s}_\theta(\boldsymbol{x}_t^i, t) + \sqrt{2\gamma_t}\boldsymbol{\epsilon}. \tag{9}$$

The next result establishes a convergence criteria on the proximity to the optimum, where for each time step $t$ there exists a minimum value of $i = \bar{I}$ such that,

$$\exists \bar{I} \text{ s.t. } \left\| (\boldsymbol{x}_t^{\bar{I}} + \gamma_t \nabla_{\boldsymbol{x}_t^{\bar{I}}} \log q(\boldsymbol{x}_t^{\bar{I}}|\boldsymbol{x}_0)) \right\|_2 \leq \|\rho_t\|_2 \tag{10}$$

where $\rho_t$ is the closest point to the global optimum that can be reached via a single gradient step from any point in $\mathbf{C}$.

**Theorem 5.2.** *Let $\mathcal{P}_{\mathbf{C}}$ be a projection onto $\mathbf{C}$, $\boldsymbol{x}_t^i$ be the sample at time step $t$ and iteration $i$, and 'Error' be the cost of the projection (7). Assume $\nabla_{\boldsymbol{x}_t} \log p(\boldsymbol{x}_t)$ is convex. For any $i \geq \bar{I}$,*

$$\mathbb{E}\left[ Error(\mathcal{U}(\boldsymbol{x}_t^i), \mathbf{C}) \right] \geq \mathbb{E}\left[ Error(\mathcal{U}(\mathcal{P}_{\mathbf{C}}(\boldsymbol{x}_t^i)), \mathbf{C}) \right] \tag{11}$$

The proof for Theorem 5.2 is reported in §H. This result suggests that PDM's projection steps ensure the resulting samples adhere more closely to the constraints as compared to samples generated

through traditional, unprojected methods. Together with the next results, it will allow us to show that PDM samples converge to the point of maximum likelihood that also satisfy the imposed constraints.

The theoretical insight provided by Theorem 5.2 provides an explanation for the observed discrepancy between the constraint violations induced by the conditional model and PDM, as in Figure 1.

**Corollary 5.3.** *For arbitrary small $\xi > 0$, there exist $t$ and $i \geq \bar{I}$ such that:*

$$Error(\mathcal{U}(\mathcal{P}_{\mathbf{C}}(\boldsymbol{x}_t^i)), \mathbf{C}) \leq \xi.$$

The above result uses the fact that the step size $\gamma_t$ is strictly decreasing and converges to zero, given sufficiently large $T$, and that the size of each update step $\mathcal{U}$ decreases with $\gamma_t$. As the step size shrinks, the gradients and noise reduce in size. Hence, $Error(\mathcal{U}(\mathcal{P}_{\mathbf{C}}(\boldsymbol{x}_t^i))$ approaches zero with $t$, as illustrated in Figure 1 (right). This diminishing error implies that the projections gradually steer the sample into the feasible subdistribution of $p(\boldsymbol{x}_0)$, effectively aligning with the specified constraints.

**Feasibility guarantees.** PDM provides feasibility guarantees when solving convex constraints. This assurance is integral in sensitive settings, such as material analysis (Section 6.1), plausible motion synthesis (Section 6.2), and physics-based simulations (Section 6.4), where strict adherence to the constraint set is necessary.

**Corollary 5.4.** *PDM provides feasibility guarantees for convex constraint sets, for* arbitrary *density functions* $\nabla_{\boldsymbol{x}_t} \log p(\boldsymbol{x}_t)$.

# 6 Experiments

We compare PDM against three methodologies, each employing state-of-the-art specialized methods tailored to the various applications tested:: **(1)** *Conditional diffusion models* (*Cond*) [13] are the state-of-the-art methods for generative sampling subject to a series of specifications. While conditional diffusion models offer a way to guide the generation process towards satisfying certain constraints, they do not provide compliance guarantees. **(2)** To encourage constraints satisfaction, we additionally compare to conditional models with a post-processing projection step (*Cond$^+$*), emulating the post-processing approaches of [10, 21] in various domains presented next. Finally, **(3)** we use a score-based model identical to our implementation but with a single post-processing projection operation (*Post$^+$*) performed at the last sampling step. Additional details are provided in §C.

The performance of these models are evaluated by the *feasibility* and *accuracy* of the generated samples. Feasibility is assessed by the degree and rate at which constraints are satisfied, expressly, the percentage of samples which satisfy the constraints with a given error tolerance. Accuracy is measured by the FID score, a standard metric in synthetic sample evaluation. To demonstrate the broad applicability of our approach, our experimental settings have been selected to exhibit:

1. Behavior in low data regimes and with original distribution violating constraints (§6.1), as part of a real-world material science experiment.
2. Behavior on 3-dimensional sequence generation with physical constraints (§6.2).
3. Behavior on complex non-convex constraints (§6.3).
4. Behavior on ODEs and under constraints outside the training distribution. (§6.4).

## 6.1 Constrained materials *(low data regimes and constraint-violating distributions)*

The first setting focuses on a *real-world* application in material science, conducted as part of an experiment to expedite the discovery of structure-property linkages (please see §C for extensive additional details). From a sparse, uniform collection of microstructure materials, we aim to generate new structures with desired, previously unobserved porosity levels.

There are two key challenges in this setting: **(1) Data sparsity:** A critical factor in this setting is the cost of producing training data. Our dataset, obtained from the authors of [5], consists of $64 \times 64$ image patches subsampled from a $3,000 \times 3,000$ pixel microscopic image, with pixel values scaled to $[-1, 1]$. These patches are upscaled to $256 \times 256$ for model training. **(2) Out-of-distribution constraints:** Constraints on the generated material's porosity, defined by pixels below a threshold representing damage, are far from those observed in the original dataset.

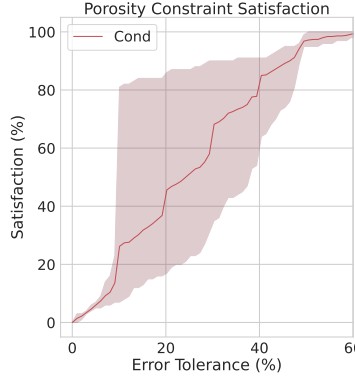

Figure 2: Conditional diffusion model (*Cond*): Frequency of porosity constraint satisfaction (y-axis) within an error tolerance (x-axis) over 100 runs.

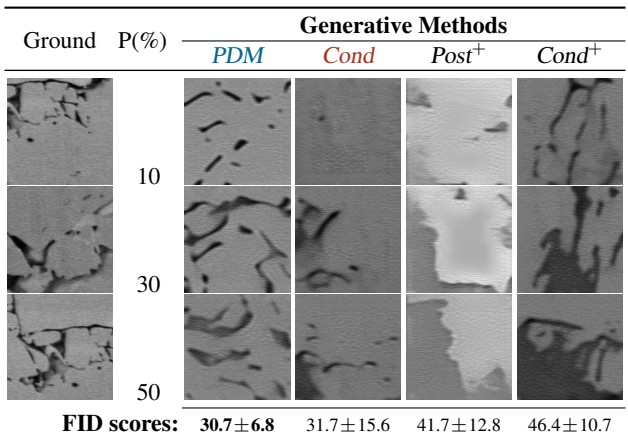

| Ground | P(%) | Generative Methods | | | |
|---|---|---|---|---|---|
| | | *PDM* | *Cond* | *Post*+ | *Cond*+ |
| | 10 | | | | |
| | 30 | | | | |
| | 50 | | | | |
| **FID scores:** | | **30.7±6.8** | 31.7±15.6 | 41.7±12.8 | 46.4±10.7 |

Figure 3: Porosity constrained microstructure visualization at varying of the imposed porosity constraint amounts (P) and FID scores.

Previous work demonstrated the use of conditional GANs [5, 11] to material generation, but these studies failed to impose verifiable constraints on desired properties. To establish a conditional baseline (denoted as *Cond*), we implement a conditional diffusion model, following the state-of-the-art approach by Chun et al. [5], conditioning the sampling on porosity measurements. Figure 2 reports the constraint violations achieved by this model. The plot depicts the frequency of constraint satisfaction (y-axis) as a function of the error tolerance (x-axis), in percentage. Observe that this state-of-the-art model struggles to adhere to the imposed constraints.

In contrast, PDM ensures both *exact* constraint satisfaction and identical image quality to the conditional model, which is significant given the complexity of the original data distribution. Figure 3 visualizes the outputs and FID scores obtained by our proposed model compared to various baselines. The constraint correction step applied by *Post*+ and *Cond*+ leads to a noticeable decrease in image quality, evident both visually and in the FID scores, rendering the generated images unsuitable for this application context. Additionally, we find that PDM outperforms *Cond* in generating microstructures that resemble those in the ground truth data (see §E.1). *These results are significant: the ability to precisely control morphological parameters in synthetic microstructures has broad impact in material synthesis, addressing critical challenges in data collection and property specification.*

### 6.2  3D human motion *(dynamic motion and physical principles)*

Next we focus on dynamic motion generation adhering to strict physical principles using the challenging HumanML3D dataset [12]. This benchmark employs three-dimensional figures across a fourth dimension of time to simulate motion. Thus, the main challenges here are generating 3D figures **(1) including a temporal component**, while **(2) ensuring they neither penetrate the floor nor float in the air**, as proposed by Yuan et al. [32]. Previous studies have seen physical law violations in motion diffusion models, with none achieving zero-tolerance results [32]. We next demonstrate PDM's capability to overcome these limitations.

First, we remark that previous approaches, such as [32], relies on a computationally demanding physics simulators to transform diffusion model predictions into "physically-plausible" actions, using a motion imitation policy trained via proximal policy optimization. This simulator is crucial to produce realistic results and dramatically alters the diffusion model's outputs. In contrast, and remarkably, PDM does not require such a simulator. and inherently satisfies the non-penetration and non-floating constraints without external assistance, showing zero violations. For comparison, the best outcomes reported in [32] ranged from 0.918 to 0.998 for penetration violations and 2.601 to 3.173 for floating violations (see §E.2 for more details).

Additionally, to more accurately assess the abilities of a diffusion model in the absences of physics simulator, we evaluate a conditional model with the same architecture as the PDM model, adapted from MotionDiffuse [33]. Results visualized in Figure 4 demonstrate that PDM achieves outputs on

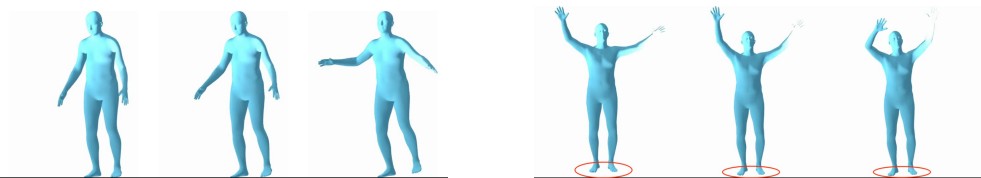

Figure 4: *PDM* (left, FID: 0.71) and conditional (*Cond*) (right, FID: 0.63) generation.

par with state-of-the-art FID scores. Additionally, *the use of projections here is guaranteed to provide exact constraint satisfaction*, as we will elaborate further in §5.

### 6.3 Constrained trajectories (*nonconvex constraints*)

The next experiment showcase the ability of PDM to handle nonconvex constraints. Path planning is a classic optimization problem which is integral to finding smooth, collision-free paths in autonomous systems. This setting consists of minimizing the path length while avoiding path intersection with various obstacles in a given topography. Recent research has demonstrated the use of diffusion models for these motion planning objectives [3]. In this task, the diffusion model predicts a series of points, $p_0, p_1, \ldots, p_N$, where each pair of consecutive points represents a line segment. The start and end points for this path are determined pseudo-randomly for each problem instance, with the topography remaining constant across different instances. Additionally, the problem presents obstacles at inference time (shown in red on Figure 5), that were not present during training, rendering a portion of the training data infeasible, and thus testing the generalization of these methods. The performance is evaluated on two sets of maps adapted from Carvalho et al., shown in Figure 5. The main challenge in this setting is the **non-convex nature of the contraints**.

To circumvent the challenge of guaranteeing collision-free paths, previous methods have relied on sampling large batches of trajectories, selecting a feasible solution if available [3]. We use the state-of-the-art *Motion Planning Diffusion* [3] as a conditional model baseline for this experiment and the associated datasets to train each of the models. For the *Cond+* model, we emulate the approach proposed by Power et al. [21], using the conditional diffusion model to generate initial points for an optimization solver. In this setting, the projection operator used by PDM is non-convex, and the implementation uses an interior point method [28]. While the feasible region is non-convex, our approach *never report unfeasible solutions* as the distance from the learned distribution decreases, unlike other methods. In contrast, using the same method for a single post-processing projection (*Post+*) does not enhance the feasibility of solutions compared to the unconstrained conditional model (*Cond*), highlighting the limitations of these single corrections in managing local infeasibilities. These observations are consistent with the analysis of Figure 1.

Figure 5 visually demonstrates that PDM can identify a feasible path with just a single sample. This capability marks a significant advance over existing state-of-the-art motion planning methods with diffusion models, as it eliminates the necessity of multiple inference attempts, which greatly affects the efficiency in generating feasible solutions. The experimental results (Table 6) demonstrate the effectiveness of PDM in handling complex, non-convex constraints in terms of success percentage for single trajectories generated (top) and path length (bottom). Notices how, both the *Cond* and *Cond+*

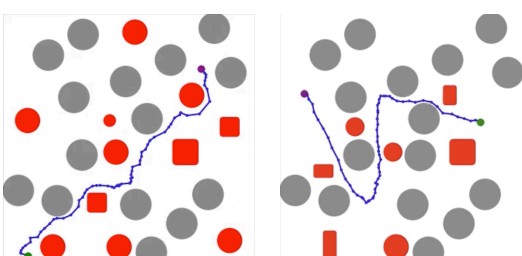

Figure 5: Constrained trajecotires synthetized by PDM on two topographies (Tp1, left and Tp2, right).

|  |  | *PDM* | *Cond* (MPD) [3] | *Cond+* |
|---|---|---|---|---|
| **S** | Tp 1 | **100.0 ± 0.0** | 77.1 ± 29.2 | 77.1 ± 29.2 |
|  | Tp 2 | **100.0 ± 0.0** | 53.3 ± 35.7 | 53.3 ± 35.7 |
| **PL** | Tp 1 | 2.21 ± 0.26 | 2.08 ± 0.51 | 2.08 ± 0.51 |
|  | Tp 2 | 2.05 ± 0.17 | 2.09 ± 0.31 | 2.09 ± 0.31 |

Figure 6: Constrained trajectories evaluation on success percentage (**S**) for a single run (higher the better, top) and path length, **PL**, (lower the better, bottom).

| t | Earth (in distribution) | | | | Moon (out of distribution) | | | |
|---|---|---|---|---|---|---|---|---|
| | *Ground* | *PDM* | *Post⁺* | *Cond⁺* | *Ground* | *PDM* | *Post⁺* | *Cond⁺* |

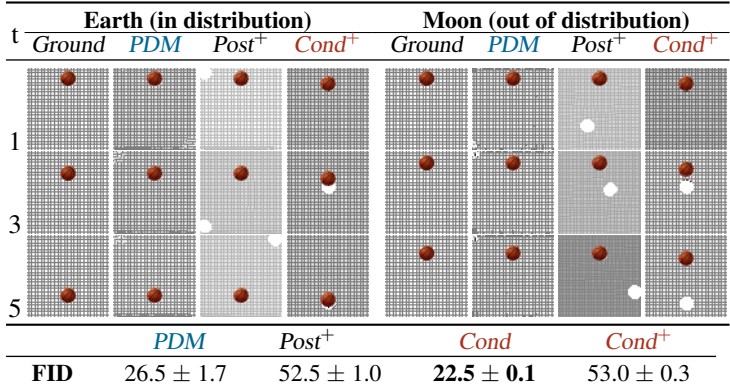

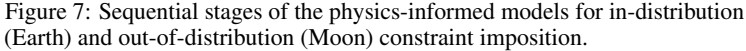

| | *PDM* | *Post⁺* | *Cond* | *Cond⁺* |
|---|---|---|---|---|
| **FID** | $26.5 \pm 1.7$ | $52.5 \pm 1.0$ | $\mathbf{22.5 \pm 0.1}$ | $53.0 \pm 0.3$ |

Figure 7: Sequential stages of the physics-informed models for in-distribution (Earth) and out-of-distribution (Moon) constraint imposition.

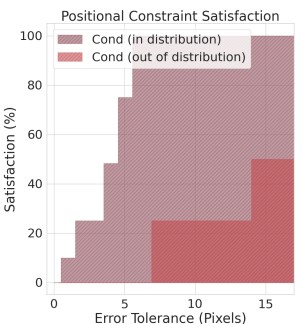

Figure 8: Conditional diffusion model (*Cond*): Frequency of constraint satisfaction (y-axis) given an error tolerance (x-axis) over 100 runs.

models fall short in finding feasible trajectories, taking shortcuts resulting in collisions. At the same time, notice how all methods find comparable path length within the reported error tolerances. *These results are significant: they show that PDM can not only handle complex, non-convex constraints, but also produce results that are on par with state-of-the-art models in solution optimality.*

### 6.4 Physics-informed motion *(ODEs and out-of-distribution constraints)*

Finally, we show the applicability of PDM in generating video frames adhering to physical principles. In this task, the goal is to generate frames depicting an object accelerating due to gravity. The object's position in a given frame is governed by

$$\mathbf{p}_t = \mathbf{p}_{t-1} + \left( \mathbf{v}_t + \left( 0.5 \times \frac{\partial \boldsymbol{v}_t}{\partial t} \right) \right) \quad (12a) \qquad \mathbf{v}_{t+1} = \frac{\partial \boldsymbol{p}_t}{\partial t} + \frac{\partial \boldsymbol{v}_t}{\partial t}, \qquad (12b)$$

where $\mathbf{p}$ is the object position, $\mathbf{v}$ is the velocity, and $t$ is the frame number. This positional information can be directly integrated into the constraint set of *PDM*, with constraint violations quantified by the pixel distance from their true position. In our experiment, the training data is based *solely* on earth's gravity and we test the model to simulate gravitational forces from the moon and other planets, in addition to earth. Thus there are two challenges in this setting **(1) satifying ODEs** describing our physical principle and **(2) generalize to out-of-distribution constraints**.

Figure 7 (left) shows randomly selected generated samples, with ground-truth images provided for reference. The subsequent rows display outputs from *PDM*, post-processing projection (*Post*), and conditional post-processing (*Cond⁺*). For this setting, we used a state-of-the-art masked conditional video diffusion model, following Voleti et al. [27]. Samples generated by conditional diffusion models are not directly shown in the figure, as the white object outline in the *Cond⁺* frames shows where the *Cond* model originally positioned the object. Notice that, without constraint projections, the score-based generative model produce samples that align with the original data arbitrarily place the object within the frame (white ball outlines in the 3rd column). Post-processing repositions the object accurately but significantly reduces image quality. Similarly, *Cond⁺* shows inaccuracies in the conditional model's object positioning, as indicated by the white outline in the 4th column. These deviations from the desired constraints are quantitatively shown in Figure 8 (light red bars), which depicts the proportion of samples adhering to the object's behavior constraints across varying error tolerance levels. Notably, this approach fails to produce *any viable sample within a zero-tolerance error margin*. In contrast, PDM generates frames that exactly satisfy the positional constraints, with FID scores comparable to those of *Cond*. Using the model proposed by Song et al. [26] further narrows this gap (see §D).

Next, Figure 7 (right) shows the behavior of the models in settings where the training data does not include any feasible data points. Here we adjust the governing equation (12) to reflect the moon's gravitational pull. Remarkably, PDM not only synthesizes high-quality images but also ensures no constraint violations (0-tolerance). This stands in contrasts to other methods, that show increased

constraint violations in out-of-distribution contexts, as shown by the dark red bars in Figure 8. *PDM can be adapted to handle complex governing equations using ODEs and can be guarantee satisfaction of out-of-distribution constraints with no decrease in sample quality.*

# 7    Discussion and limitations

In many scientific and engineering domains and safety-critical applications, constraint satisfaction guarantees are a critical requirement. It is however important to acknowledge the existence of an inherent trade-off, particularly in computational overhead. In applications where inference time is a critical factor, it may be practical to adjust the time step $t$ at which iterative projections begin, which guides a trade-off between the FID score associated with the starting point of iterative projections and the computational cost of projecting throughout the remaining iterations (§F). Other avenues to improve efficiency also exists, from the adoption of specialized solvers within the application domain of interest to the adoption of warm-start strategies for iterative solvers. The latter, in particular, relies exploiting solutions computed in previous iterations of the sampling step and was found to be a practical strategy to substantially decrease the projections overhead.

We also note the absence of constraints in the forward process. As illustrated empirically, it is unnecessary for the training data to contain any feasible points. We hold that this not only applies to the final distribution but to the interim distributions as well. Furthermore, by projecting perturbed samples, the cost of the projection results in divergence from the distribution that is being learned. Hence, we conjecture that incorporating constraints into the forward process will not only increase computational cost of model training but also decrease the FID scores of the generated samples.

Finally, while this study provides a framework for imposing constraints on diffusion models, the representation of complex constraints for multi-task large scale models remains an open challenge. This paper motivates future work for adapting optimization techniques to such settings, where constraints ensuring accuracy in task completion and safety in model outputs bear transformative potential to broaden the application of generative models in many scientific and engineering fields.

# 8    Conclusions

This paper was motivated by a significant challenge in the application of diffusion models in contexts requiring strict adherence to constraints and physical principles. It presented Projected Diffusion Models (PDM), an approach that recasts the score-based diffusion sampling process as a constrained optimization process that can be solved via the application of repeated projections. Experiments in domains ranging from physical-informed motion for video generation governed by ordinary differentiable equations, trajectory optimization in motion planning, and adherence to morphometric properties in generative material science processes illustrate the ability of PDM to generate content of high-fidelity that also adheres to complex non-convex constraints as well as physical principles.

# 9    Acknowledgments

This research is partially supported by NSF grants 2334936, 2334448, and NSF CAREER Award 2401285. Fioretto is also supported by an Amazon Research Award and a Google Research Scholar Award. The authors acknowledge Research Computing at the University of Virginia for providing computational resources that have contributed to the results reported within this paper. The views and conclusions of this work are those of the authors only.

**Authors Contributions**

JC and FF formulated the research question, designed the methodology, developed the theoretical analysis, and wrote the manuscript. Moreover, JC contributed to developing the code and performed the experimental analysis. SB acquired the data for the micro-structure experiment, formulated the desired properties for such experiment, and participated in the interpretation of the results.

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

# A Broader impacts

The development of Projected Diffusion Models (PDM) may significantly enhance the application of diffusion models in fields requiring strict adherence to specific constraints and physical principles. The proposed method enables the generation of high-fidelity content that not only resembles real-world data but also complies with complex constraints, including non-convex and physical-based specifications. PDM's ability to handle diverse and challenging constraints in scientific and engineering domains, particularly in low data environments, may potentially lead to accelerating innovation and discovery in various fields.

# B Expanded related work

**Diffusion models with soft constraint conditioning.** Variations of conditional diffusion models [13] serve as useful tools for controlling task specific outputs from generative models. These methods have demonstrated the capacity capture properties of physical design [29], positional awareness [3], and motion dynamics [32] through augmentation of these models. The properties imposed in these architectures can be viewed as soft constraints, with stochastic model outputs violating these loosely imposed boundaries.

**Post-processing optimization.** In settings where hard constraints are needed to provide meaningful samples, diffusion model outputs have been used as starting points for a constrained optimization algorithm. This has been explored in non-convex settings, where the starting point plays an important role in whether the optimization solver will converge to a feasible solution [21]. Other approaches have augmented the diffusion model training objective to encourage the sampling process to emulate an optimization algorithm, framing the post-processing steps as an extension of the model [10, 19]. However, an existing challenge in these approaches is the reliance on an easily expressible objective, making these approaches effective in a limited set of problems (such as the constrained trajectory experiment) while not applicable for the majority of generative tasks.

**Hard constraints for generative models.** Frerix et al. [9] proposed an approach for implementing hard constraints on the outputs of autoencoders. This was achieved through scaling the generated outputs in such a way that feasibility was enforced, but the approach is to limited simple linear constraints. [17] proposed an approach to imposing constraints using "mirror mappings" with applicability exclusively to common, convex constraint sets. Due to the complexity of the constraints imposed in this paper, neither of these methods were applicable to the constraint sets explored in any of the experiments. Alternatively, work by Fishman et al. [2023, 2024] broadens the classes of constraints that can be represented but fails to demonstrate the applicability of their approach to a empirical settings similar to ours, utilizing an MLP architecture for trivial predictive tasks with constraints sets that can be represented by convex polytopes. We contrast such approaches to our work, noting that this prior work is limited to constraint sets that can be approximated by simple neighborhoods, such as an L2-ball, simplex, or polytope, whereas PDM can handle constraint sets of arbitrary complexity.

**Sampling process augmentation.** Motivated by the compounding of numerical error throughout the reverse diffusion process, prior work has proposed inference time operations to bound the pixel values of an image dynamically while sampling [18, 23]. Proposed methodologies have either applied reflections or simple clipping operations during the sampling process, preventing the generated image from significantly deviating from the [0,255] pixel space. Such approaches augment the sampling process in a way that mirrors our work, but these methods are solely applicable to mitigating sample drift and do not intersect our work in general constraint satisfaction.

# C Experimental settings

In the following section, further details are provided as to the implementations of the experimental settings used in this paper.

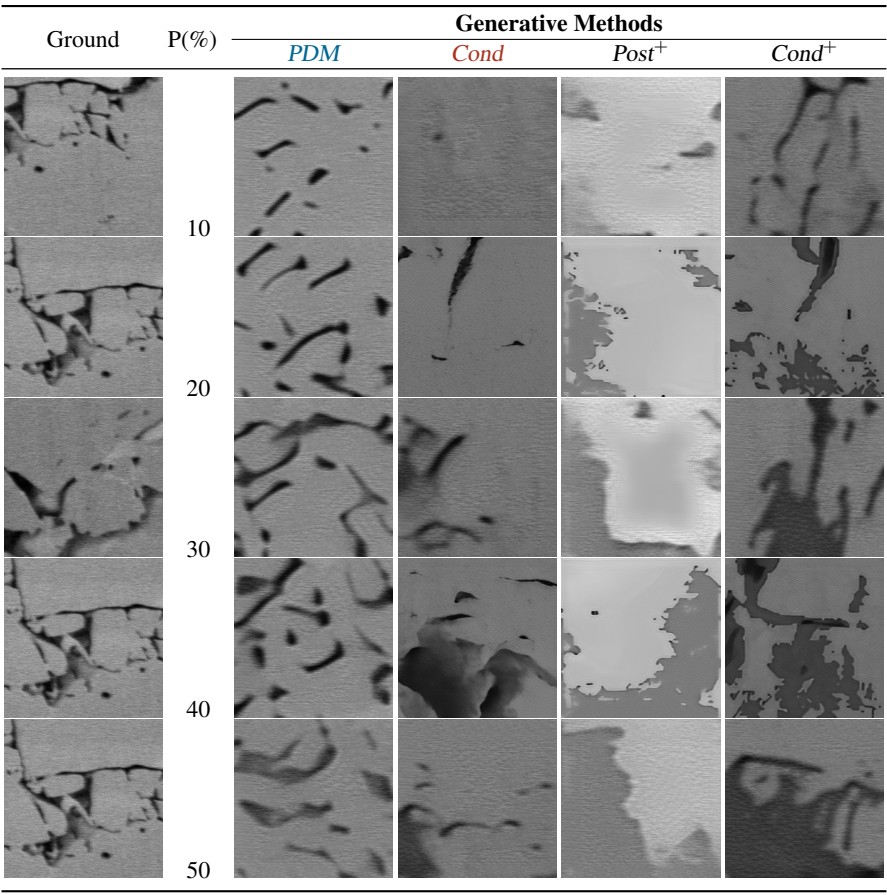

Figure 9: Porosity constrained microstructure visualization at varying of the imposed porosity constraint amounts (expanded from Figure 3).

## C.1 Constrained materials

Microstructures are pivotal in determining material properties. Current practice relies on physics-based simulations conducted upon imaged microstructures to quantify intricate structure-property linkages [4]. However, acquiring real material microstructure images is both costly and time-consuming, lacking control over attributes like porosity, crystal sizes, and volume fraction, thus necessitating "cut-and-try" experiments. Hence, the capability to generate realistic synthetic material microstructures with controlled morphological parameters can significantly expedite the discovery of structure-property linkages.

Previous work has shown that conditional generative adversarial networks (GAN) [11] can be used for this end [5], but these studies have been unable to impose verifiable constraints on the satisfaction of these desired properties. To provide a conditional baseline, we implement a conditional DDPM modeled after the conditional GAN used by Chun et al. [5] with porosity measurements used to condition the sampling.

**Projections.** The porosity of an image is represented by the number of pixels in the image which are classified as damaged regions of the microstructure. Provided that the image pixel intensities are scaled to [-1, 1], a threshold is set at zero, with pixel intensities below this threshold being classified as damage regions. To project, we implement a top-k algorithm that leaves the lowest and highest intensity pixels unchanged, while adjusting the pixels nearest to the threshold such that the total number of pixels below the threshold precisely satisfies the constraint.

**Conditioning.** The conditional baseline is conditioned on the porosity values of the training samples. The implementation of this model is as described by Ho and Salimans.

| t | Earth (in distribution) | | | | Moon (out of distribution) | | | |
|---|---|---|---|---|---|---|---|---|
| | *Ground* | *PDM* | *Post*[+] | *Cond*[+] | *Ground* | *PDM* | *Post*[+] | *Cond*[+] |

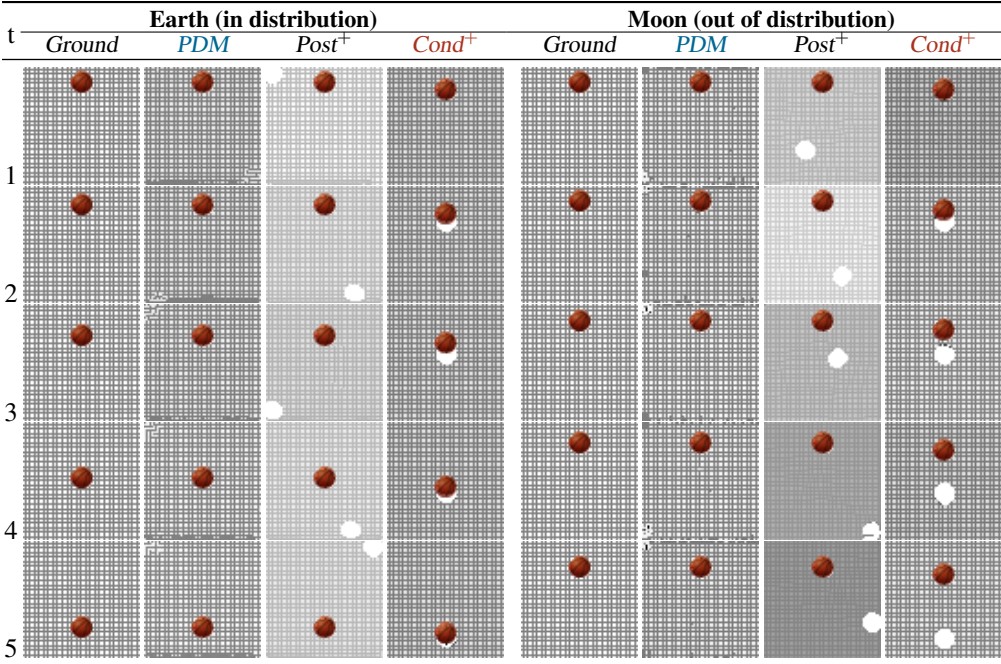

Figure 10: Sequential stages of the physics-informed models for in-distribution (Earth) and out-of-distribution (Moon) constraint imposition (expanded from Figure 7).

**Original training data.** We include samples from the original training data to visually illustrate how closely our results perform compared to the real images. As the specific porosities we tested on are not adhered to in the dataset, we illustrate this here as opposed to in the body of the text.

We observe that only the Conditional model and PDM synthesize images that visually adhere to the distribution, while post-processing methods do not provide adequate results for this complex setting.

## C.2   3D human motion

**Projections.** The penetration and floatation constraints can be handled by ensuring that the lowest point on the z-axis is equal to the floor height. Additionally, to control the realism of the generated figures, we impose equality constraints on the size of various body parts, including the lengths of the torso and appendages. These constraints can be implemented directly through projection operators.

**Conditioning.** The model is directly conditioned on text captions from the HumanML3D dataset. The implementation is as described in [33].

## C.3   Constrained trajectories

**Projections.** For this experiment, we represent constraints such that the predicted path avoids intersecting the obstacles present in the topography. These are parameterized to a non-convex interior point method solver. For circular obstacles, this can be represented by a minimum distance requirement, the circle radius, imposed on the nearest point to the center falling on a line between $p_n$ and $p_{n+1}$. These constraints are imposed for all line segments. We adapt a similar approach for non-circular obstacles by composing these of multiple circular constraints, hence, avoiding over-constraining the problem. More customized constraints could be implement to better represent the feasible region, likely resulting in shorter path lengths, but these were not explored for this paper.

**Conditioning.** The positioning of the obstacles in the topography are passed into the model as a vector when conditioning the model for sampling. Further details can be found the work presented by Carvalho et al., from which this baseline was directly adapted.

## C.4 Physics-informed motion

The dataset is generated with object starting points sampled uniformly in the interval [0, 63]. For each data point, six frames are included with the position changing as defined in Equation 12 and the initial velocity $\mathbf{v}_0 = 0$. Pixel values are scaled to [-1, 1]. The diffusion models are trained on 1000 points with a 90/10 train/test split.

**Projections.** Projecting onto positional constraints requires a two-step process. First, the current position of the object is identified and all the pixels that make up the object are set to the highest pixel intensity (white), removing the object from the original position. The set of pixel indices representing the original object structure are stored for the subsequent step. Next, the object is moved to the correct position, as computed by the constraints, as each pixel from the original structure is placed onto the center point of the true position. Hence, when the frame is feasible prior to the projection, the image is returned unchanged, which is consistent with the definition of a projection.

**Conditioning.** For this setting, the conditional video diffusion model takes two ground truth frames as inputs, from which it infers the trajectory of the object and the starting position. The model architecture is otherwise as specified by Voleti et al..

# D    PDM for score-based generative modeling through stochastic differential equations

## D.1    Algorithms

While the majority of our analysis focused on the developing these techniques to the sampling architecture proposed for Noise Conditioned Score Networks [25], this approach can directly be adapted to the diffusion model variant Score-Based Generative Modeling with Stochastic Differential Equations proposed by Song et al. Although our observations suggested that optimizing across a continuum of distributions resulted in less stability in diverse experimental settings, we find that this method is still effective in producing high-quality constrained samples in others.

We included an updated version of Algorithm 1 adapted to these architectures.

---

**Algorithm 2:** PDM Corrector Algorithm

---

1  $\mathbf{x}_N^0 \sim \mathcal{N}(\mathbf{0}, \sigma_{\max}^2 \mathbf{I})$
2  **for** $t \leftarrow T$ *to 1* **do**
3      **for** $i \leftarrow 1$ *to* $M$ **do**
4         $\epsilon \sim \mathcal{N}(\mathbf{0}, \mathbf{I})$
5         $\mathbf{g} \leftarrow \mathbf{s}_{\theta*}(\mathbf{x}_t^{i-1}, \sigma_t)$
6         $\gamma \leftarrow 2(r||\epsilon||_2/||\mathbf{g}||_2)^2$
7         $\mathbf{x}_t^i \leftarrow \mathcal{P}_{\mathbf{C}}(\mathbf{x}_t^{i-1} + \gamma\mathbf{g} + \sqrt{2\gamma}\epsilon)$
8      $\mathbf{x}_{t-1}^0 \leftarrow \mathbf{x}_t^M$
9  **return** $\mathrm{x}_0^0$

---

We note that a primary discrepancy between this algorithm and the one presented in Section 4.2 is the difference in $\gamma$. As the step size is not strictly decreasing, the guidance effect provided by PDM is impacted as Corollary 5.3 does not hold for this approach. Hence, we do not focus on this architecture for our primary analysis, instead providing supplementary results in the subsequent section.

## D.2    Results

We provide additional results using the Score-Based Generative Modeling with Stochastic Differential Equations. This model produced highly performative results for the Physics-informed Motion experiment, with visualisations included in Figures 11 and 12. This model averages an impressive inception score of **24.2** on this experiment, slightly outperforming the PDM implementation for Noise

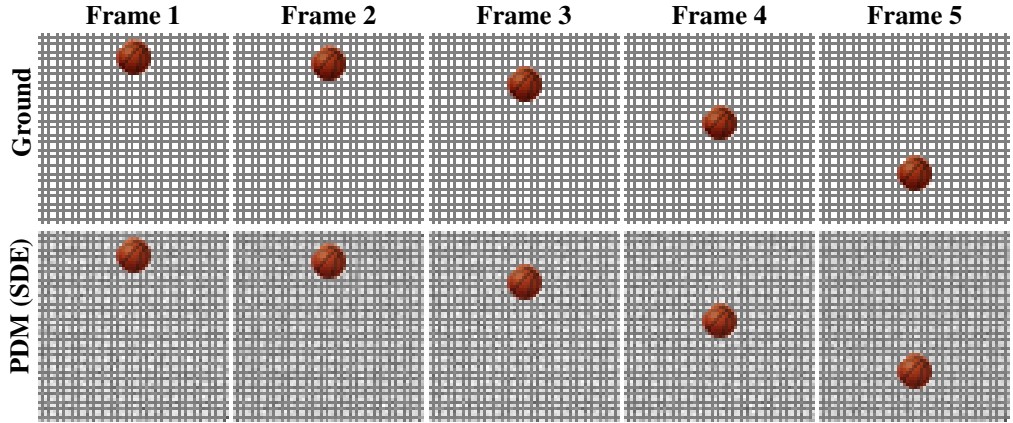

Figure 11: In distribution sampling for physics-informed model via Score-Based Generative Modeling with SDEs.

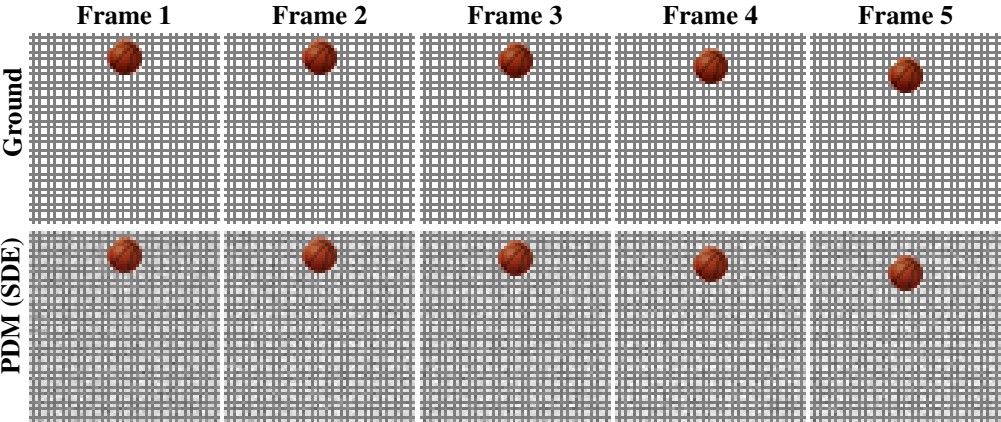

Figure 12: Out of distribution sampling for physics-informed model via Score-Based Generative Modeling with SDEs.

Conditioned Score Networks. Furthermore, it is equally capable in generalizing to constraints that were not present in the training distribution.

# E    Additional results

## E.1    Constrained materials morphometric parameter distributions

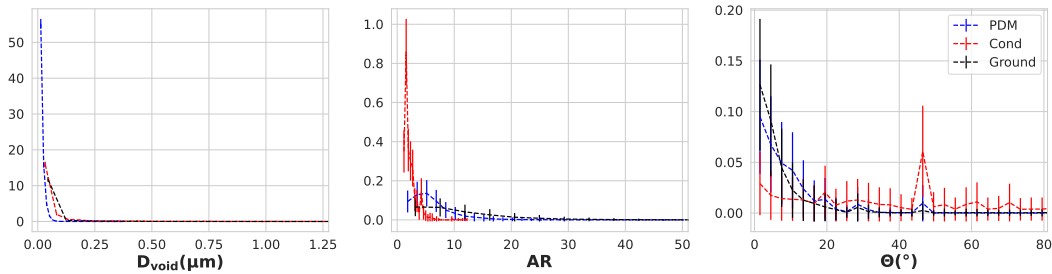

Figure 13: Distributions of the morphometric parameters, comparing the ground truth to *PDM* and *Cond* models using heuristic-based analysis.

When analyzing both real and synthetic materials, heuristic-guided metrics are often employed to extract information about microstrucutres present in the material. When analyzing the quality of synthetic samples, the extracted data can then be used to assess how well the crystals and voids in the microstructure adhere to the training data, providing an additional qualitative metrics for analysis. To augment the metrics displayed within the body of the paper, we include here the distribution of three metrics describing these microstructures, mirroring those used by Chun et al..

We observe that the constraint imposition present in PDM improves the general adherence of the results to the ground truth microstructures. This suggests that the *Cond* model tends to generate to certain microstructures at a frequency that is not reflected in the training data. By imposing various porosity constraints, PDM is able to generate a more representative set of microstructures in the sampling process.

## E.2 3D human motion

We highlight that unlike the approach proposed by [32], our approach guarantees the generated motion does not violate the penetrate and float constraints. The results are tabulated in Table 1 (left) and report the violations in terms of measured distance the figure is either below (penetrate) or above (float) the floor. For comparison, we include the projection schedules utilized by PhysDiff which report the best results to show that even in these cases the model exhibits error.

| Method | FID | Penetrate | Float |
|---|---|---|---|
| PhysDiff [32] (Start 3, End 1) | 0.51 | 0.918 | 3.173 |
| PhysDiff [32] (End 4, Space 1) | **0.43** | 0.998 | 2.601 |
| *PDM* | 0.71 | **0.00** | **0.00** |

Table 1: PDM performance compared to (best) PhysDiff results on HumanML3D.

The implementation described by [32] applies a physics simulator at scheduled intervals during the sampling process to map the diffusion model's prediction to a "physically-plausible" action that imitates data points of the training distribution. This simulator dramatically alters the diffusion model's outputs utilizing a learned motion imitation policy, which has been trained to match the ground truth samples using proximal policy optimization. In this setting the diffusion model provides a starting point for the physics simulator and is not directly responsible for the final results of these predictions. Direct parallels can be drawn between this approach and other methods which solely task the diffusion model with initializing an external model [10, 21]. Additionally, while the authors characterize this mapping as a projection, it is critical to note that this is a projection onto the learned distribution of the simulator and not a projection onto a feasible set, explaining the remaining constraint violations in the outputs.

## E.3 Convergence of PDM

As shown in Figure 1, the PDM sampling process converges to a feasible subdistribution, a behavior that is generally not present in standard conditional models. Corollary 5.3 provides insight into this behavior as it outlines the decreasing upper bound on *'Error'* that can be introduced in a single sampling step. To further illustrate this behavior, the decreasing upper bound can be illustrated in Figure 14.

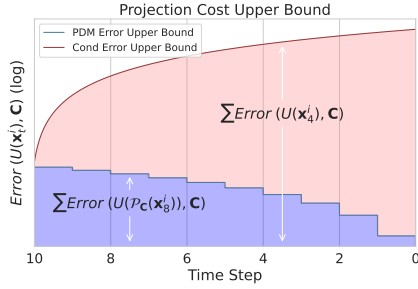

Figure 14: Visualization of the decreasing upper bound on error introduced in a single sampling step for *PDM*, as opposed to the strictly increasing upper bound of conditional (*Cond*) models.

## F Computational costs

To compare the computational costs of sampling with PDM to our baselines, we record the execution times for the reverse process of a single sample. *The implementations of PDM have not been optimized for runtime, and represent an upper bound.* All sampling is run on two NVIDIA A100 GPUs. All computations are conducted on these GPUs with

the exception of the interior point method projection used in the 3D Human motion experiment and the Constrained Trajectories experiment which runs on two CPU cores.

| | Constrained Materials | 3D Human Motion | Constrained Trajectories | Physics-informed Motion |
|---|---|---|---|---|
| *PDM* | 26.89 | 682.40* | 383.40* | 48.85 |
| *Post⁺* | 26.01 | – | – | 27.58 |
| *Cond* | 18.51 | 13.79 | 0.56 | 35.30 |
| *Cond⁺* | 18.54 | – | 106.41 | 36.63 |

Table 2: Average sampling run-time in seconds.

We implement projections at all time steps in this analysis, although practically this is can be optimized to reduce the total number of projections as described in the subsequent section. Additionally, we set $M = 100$ and $T = 10$ for each experiment. The increase in computational cost present in PDM is directly dependant on the tractability of the projections and the size of $M$.

The computational cost of the projections is largely problem dependant, and we conjecture that these times could be improved by implementing more efficient projections. For example, the projection for Constrained Trajectories could be dramatically improved by implementing this method on the GPUs instead of CPUs (*). However, these improvements are beyond the scope of this paper. Our projection implementations are further described in §C.

Additionally, the number of iterations for each $t$ can often be decreased below $M = 100$ or the projection frequency can be adjusted (as has been done for in this section for the CPU implemented projections), offering additional speed-up.

# G   Variational lower bound training objective

As defined in Equation 2, PDM uses a score-matching objective to learn to the gradients of the log probability of the data distribution. This understanding allows the sampling process to be framed in a light that is consistent to optimization theory, allowing equivalences to be drawn between the proposed sampling procedure and projected gradient descent. This aspect is also integral to the theory presented in Section 4.2.

Other DDPM and DDIM implementation utilize a variation lower bound objective, which is a tractable approach to minimizing the negative log likelihood on the network's noise predictions. While this approach was inspired by the score-matching objective, we empirically demonstrate that iterative projections perform much worse in our tested settings than models optimized using this training

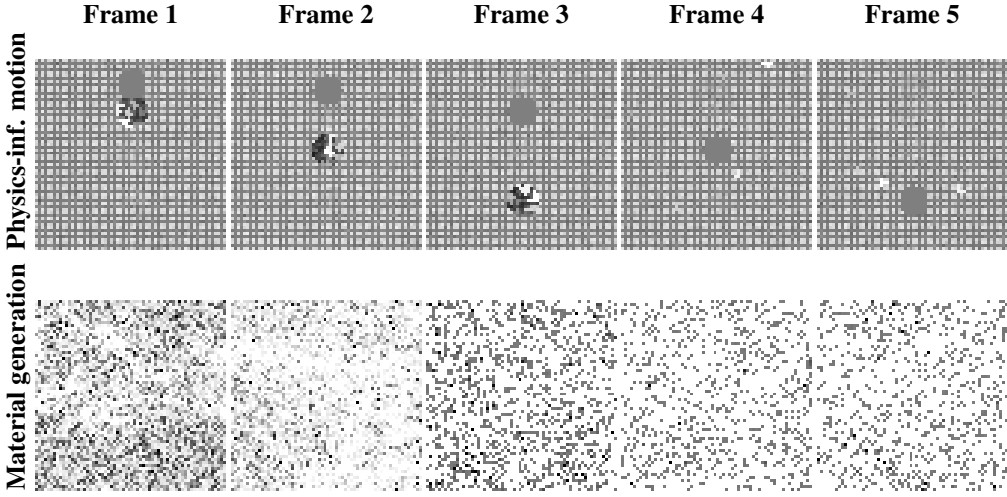

Figure 15: Iterative projections using model trained with variational lower bound objective.

objective, producing clearly inferior solutions in the Physics-informed experiments and failing to produce viable solutions in the material science domain explored.

This approach (visualized in Figure 15) resulted in an FID score of $113.8 \pm 4.9$ on the Physics-informed Motion experiment and $388.2 \pm 13.0$ on the Constrained Materials experiment, much higher than those produced using the score-matching objective, adopted in our paper. We hold that this is because the approach proposed in our paper is more theoretically sound when framed in terms of a gradient-based sampling process.

# H    Missing proofs

**Proof of Theorem 5.2**

*Proof.* By optimization theory of convergence in a convex setting, provided an arbitrarily large number of update steps $M$, $\boldsymbol{x}_t^M$ will reach the global minimum. Hence, this justifies the existence of $\bar{I}$ as at some iteration as $i \rightarrow \infty$, $\left\|\boldsymbol{x}_t^i + \gamma_t \nabla_{\boldsymbol{x}_t^i} \log q(\boldsymbol{x}_t^i|\boldsymbol{x}_0)\right\|_2 \leq \|\rho_t\|_2$ will hold for every iteration thereafter.

Consider that a gradient step is taken without the addition of noise, and $i \geq \bar{I}$. Provided this, there are two possible cases.

**Case 1:**    Assume $\boldsymbol{x}_t^i + \gamma_t \nabla_{\boldsymbol{x}_t^i} \log q(\boldsymbol{x}_t^i|\boldsymbol{x}_0)$ is closer to the optimum than $\rho_t$. Then, $\boldsymbol{x}_t^i$ is infeasible.

This claim is true by the definition of $\rho_t$, as $\boldsymbol{x}_t^i + \gamma_t \nabla_{\boldsymbol{x}_t^i} \log q(\boldsymbol{x}_t^i|\boldsymbol{x}_0)$ is closer to $\mu$ than is achievable from the nearest feasible point to $\mu$. Hence, $\boldsymbol{x}_t^i$ must be infeasible.

Furthermore, the additional gradient step produces a point that is closer to the optimum than possible by a single update step from the feasible region. Hence it holds that

$$Error(\boldsymbol{x}_t^i + \gamma_t \nabla_{\boldsymbol{x}_t^i} \log q(\boldsymbol{x}_t^i|\boldsymbol{x}_0)) > Error(\mathcal{P}_{\mathbf{C}}(\boldsymbol{x}_t^i) + \gamma_t \nabla_{\mathcal{P}_{\mathbf{C}}(\boldsymbol{x}_t^i)} \log q(\mathcal{P}_{\mathbf{C}}(\boldsymbol{x}_t^i)|\boldsymbol{x}_0)) \qquad (13)$$

as the distance from the feasible region to the projected point will be at most the distance to $\rho_t$. As this point is closer to the global optimum than $\rho_t$, the cost of projecting $\boldsymbol{x}_t^i + \gamma_t \nabla_{\boldsymbol{x}_t^i} \log q(\boldsymbol{x}_t^i|\boldsymbol{x}_0)$ is greater than that of any point that begins in the feasible region.

**Case 2:**    Assume $\boldsymbol{x}_t^i + \gamma_t \nabla_{\boldsymbol{x}_t^i} \log q(\boldsymbol{x}_t^i|\boldsymbol{x}_0)$ is equally close to the optimum as $\rho_t$. In this case, there are two possibilities; either (1) $\boldsymbol{x}_t^i$ is the closest point in $\mathbf{C}$ to $\mu$ or (2) $\boldsymbol{x}_t^i$ is infeasible.

If the former is true, $\boldsymbol{x}_t^i = \mathcal{P}_{\mathbf{C}}(\boldsymbol{x}_t^i)$, implying

$$Error(\boldsymbol{x}_t^i + \gamma_t \nabla_{\boldsymbol{x}_t^i} \log q(\boldsymbol{x}_t^i|\boldsymbol{x}_0)) = Error(\mathcal{P}_{\mathbf{C}}(\boldsymbol{x}_t^i) + \gamma_t \nabla_{\mathcal{P}_{\mathbf{C}}(\boldsymbol{x}_t^i)} \log q(\mathcal{P}_{\mathbf{C}}(\boldsymbol{x}_t^i)|\boldsymbol{x}_0)) \qquad (14)$$

Next, consider that the latter is true. If $\boldsymbol{x}_t^i$ is not the closest point in $\mathbf{C}$ to the global minimum, then it must be an equally close point to $\mu$ that falls outside the feasible region. Now, a subsequent gradient step of $\boldsymbol{x}_t^i$ will be the same length as a gradient step from the closest feasible point to $\mu$, by our assumption.

Since the feasible region and the objective function are convex, this forms a triangle inequality, such that the cost of this projection is greater than the size of the gradient step. Thus, by this inequality, Equation 13 applies.

Finally, for both cases we must consider the addition of stochastic noise. As this noise is sampled from the Gaussian with a mean of zero, we synthesize this update step as the expectation over,

$$\mathbb{E}\left[Error(\boldsymbol{x}_t^i + \gamma_t \nabla_{\boldsymbol{x}_t^i} \log q(\boldsymbol{x}_t^i|\boldsymbol{x}_0) + \sqrt{2\gamma_t}\boldsymbol{\epsilon})\right] \geq \mathbb{E}\left[Error(\mathcal{P}_{\mathbf{C}}(\boldsymbol{x}_t^i) + \gamma_t \nabla_{\mathcal{P}_{\mathbf{C}}(\boldsymbol{x}_t^i)} \log q(\mathcal{P}_{\mathbf{C}}(\boldsymbol{x}_t^i)|\boldsymbol{x}_0) + \sqrt{2\gamma_t}\boldsymbol{\epsilon})\right]$$
$$(15)$$

or equivalently as represented in Equation 11.

$\square$

