# OpenReview forum: "Constrained Synthesis with Projected Diffusion Models"
_NeurIPS.cc/2024/Conference — NeurIPS 2024 poster_

### Official Review · Reviewer_p9zr · 2024-06-26

**Soundness:** 4
**Presentation:** 4
**Contribution:** 3
**Rating:** 7
**Confidence:** 3

**Summary:**

This paper proposes an approach to sample generation using diffusion models which adheres to a set of constraints. The approach is based on the score matching formulation of diffusion models, and applies a projection step which finds the nearest feasible sample to each iteration of SGLD. A theoretical justification for the method is proposed, which provides intuition around why the approach yields superior results to conditioning and post-processing methods. The method is evaluated on a wide variety of experiments covering multiple classes of constraints (including non-convex) and data modalities, and is demonstrated to yield high-quality samples which strictly adhere to non-trivial constraints.

**Strengths:**

This is a very strong submission which is well-written and easy to read. The method is very sensible, and described clearly.

The experiments are diverse and interesting, spanning multiple application domains, data modalities (e.g., images, configuration space trajectories) and constraint types. In particular, the motion planning experiments with non-convex constraints are particularly interesting to me since synthesis with non-convex constraints has not really been approached in prior works.

Furthermore, the demonstration of synthesis with constraints applied only at test time, with the constraints not being applied during training and having the training data violating the constraints is also very appealing.

Finally, the authors also provide some theoretical justification and intuition of the proposed approach for convex constraints, which is a nice addition.

**Weaknesses:**

I think a mathematical description of the projection operator and constraints for the experiments 5.1-5.3 would be helpful to understand the degree of non-convexity and difficulty for these experiments. While I have a rough idea, being concrete would help a lot.

Also, I believe it is important to be more forward in the main document around the computational overhead when it comes to applying the projection operator at each SGLD iteration. While I believe that high overhead does not particularly detract from the contribution (this can be addressed in future works), I think it detracts from the paper to hide this information in the appendix and not refer to it obviously in the main text.

**Questions:**

line 158: Does "convergence of convex constraint sets" simply mean that the projection step will always return a feasible solution? Or are you referring more broadly to the DDPM iterates?
line 186: Double colon typo "::"
Fig. 1: Given that the approach projects to the constraint set at each iteration, why are we still seeing constraint violation early on in the diffusion process? Is this plotting the pre-projected samples? It is unclear what is actually being plotted.
line 272: typo, "unfeasible"
line 331: some comment around the reasonability of the assumption of the score function being convex (locally for small step size say?) would be appreciated

**Limitations:**

The authors have clearly described the limitations of the approach, however I believe a clearer statement of compute overhead would improve the clarity around the limitations.

---

> ### Author Rebuttal · Authors · 2024-08-03
>
> Thank you for their time reviewing our work and their praise of our submission. We appreciate your consideration of our work and would like to address your outstanding questions and concerns.
>
> >**Weakness 1: I think a mathematical description of the projection operator and constraints for the experiments 5.1-5.3 would be helpful to understand the degree of non-convexity and difficulty for these experiments. While I have a rough idea, being concrete would help a lot.**
>
> - **Constrained Materials:** For this experiment we use a classical knapsack constraint. Note that while in general knapsack problems are NP-complete, the version adopted (which uses integers as weight) is known to be weakly NP-complete and admits a fully polynomial approximation. It is solved efficiently in $O(nm)$ where $n$ is the number of pixels and $m$ is the number of values to compute for the dynamic program.
> - **3D Human Motion:** This is a scaling constraint that adjusts all positions of the figure by the relative distance between the lowest joint and the floor, adjusting the relative position to prevent floating and penetration i.e.,
> $ argmin ||y-x||$ s.t. $\min_i (y_i) = 0 \quad \text{and} \quad ||y_j - y_k|| = ||x_j - x_k|| \forall j, k$.
> Additionally, we impose a realism constraint on appendages, keeping them consistent with one another i.e.
> $||elbow_\textrm{right} - wrist_\textrm{right}|| = ||elbow_\textrm{left} - wrist_\textrm{left}||$. The implementation runs in $O(n)$ time, where $n$ is the length of the internal representation.
> - **Constrained Trajectories:** This problem primarily represents constraints as a minimum distance, $d$, between the center point of an obstacle, $q$, and the closest point falling between each set of consecutive points. More formally, this is expressed by evaluating the distance $d_{\text{min}}$ from $q$ to the line segment $\overline{p_i p_{i+1}}$, where $d_{\text{min}}$ is determined as follows: if the projection of $q$ onto the line defined by $p_i$ and $p_{i+1}$ falls within the segment, $d_{\text{min}}$ is the perpendicular distance from $q$ to the line; otherwise, $d_{\text{min}}$ is the distance from $q$ to the nearest endpoint, $p_i$ or $p_{i+1}$. The constraint is thus $d_{\text{min}} > d$, ensuring that the nearest point on the segment to $q$ is at least a distance $d$ away.
> The interior point method used by the nonconvex solver in our implementation [27] has a time complexity of $O(n^{3.5})$ where $n$ is the number of variables in the Quadratic Program. In this case, there are 128.
> - **Physics-informed Motion:** This is similar to Section 5.2 and runs in $O(n+m)$ where the size of each image is $n$ by $m$.
>
>
> We appreciate this suggestion and will add the formalization of these operators to our next draft.
>
> >**Weakness 2: Also, I believe it is important to be more forward in the main document around the computational overhead when it comes to applying the projection operator at each SGLD iteration. While I believe that high overhead does not particularly detract from the contribution (this can be addressed in future works), I think it detracts from the paper to hide this information in the appendix and not refer to it obviously in the main text.**
>
> We are open to additional suggestions on how to more obviously showcase this component in our paper. While we do explicitly reference Section F in our Discussion and limitations section (Section 7), we were unable to make space to display this table in the nine pages of the main paper allowed for submission. This is definetively something we will address in the extra page allowed for the accepted version.
>
> >**Question 1: line 158: Does "convergence of convex constraint sets" simply mean that the projection step will always return a feasible solution? Or are you referring more broadly to the DDPM iterates?**
>
> This line specifically refers to the latter, as **by proximal gradient descent theory we can guarantee convergence to both a feasible and optimal solution when the problem is convex.** You are also correct in that PDM _guarantees_ feasiblity for convex constraints!
>
> >**Question 2: Fig. 1: Given that the approach projects to the constraint set at each iteration, why are we still seeing constraint violation early on in the diffusion process? Is this plotting the pre-projected samples? It is unclear what is actually being plotted.**
>
> Figure 1 shows the samples after the gradient step is applied (line 6 of Algorithm 1) but **prior to the projection.** The point we would like to illustrate here is that later, in the reverse process, the projections have minimal impact on the sample as additional diffusion steps do not result in constraint violations. This also means that the samples do not directly fall on the constraint boundaries as they converge within the feasible region. Conversely, when post-processing a conditional model's outputs (*Cond+*), the sample is dramatically altered by this projection, resulting in _much higher FID scores_ and _deviation from the real data distribution_.
>
> ---
> We appreciate your review and feedback and are happy to address any other questions you may have. It is our hope that these responses have addressed any concerns that you may have and provided confidence in supporting even further our work! Thank you for your time and consideration.

---

> ### Author Response · Authors · 2024-08-10
>
> We thank Reviewer p9zr for their thoughtful feedback and for recognizing the strengths of our submission and praising our experiments as _diverse and interesting_. We appreciate the opportunity to clarify the aspects of our work that you highlighted in your review.
> In our main rebuttal, we addressed the following key points:
> 1. **Detailed Mathematical Descriptions**: We have provided a more detailed mathematical description of the projection operators used in experiments 5.1-5.3. This includes the specific constraints applied and their computational complexities, which help illustrate the degree of non-convexity and the challenges associated with these experiments.
> 2. **Computational Overhead**: We acknowledge your point and this information is now more prominently discussed.
> 3. **Other Clarifications**: For Figure 1: it shows the samples post-gradient step but pre-projection. This helps in understanding the iterative convergence of the method within the constraint set through the diffusion process. Additionally, we provided a clearer explanation of what we mean by “convergence of convex constraint sets” in line 158, linking it to both feasibility and optimality in the context of proximal gradient descent.
>
> Are there any additional concerns that we could address? We are ready to provide further insights to assist in your evaluation and to enhance the understanding of our findings.

---

> > ### Author Response · Authors · 2024-08-11
> >
> > As the discussion period is nearing its end, we wanted to ask if there are any follow-up points we can clarify. Please also note our summary in the previous comment. Many thanks!

---

> > > ### Comment · Reviewer_p9zr · 2024-08-12
> > > **Response to the authors**
> > >
> > > Thank you for your detailed response to my comments. This has addressed my comments, and I am glad the authors' will make the necessary changes to the manuscript. My opinion of the paper is still positive and remains unchanged by the discussion period.

---

> > > > ### Author Response · Authors · 2024-08-13
> > > >
> > > > Thank you very much for your recognition of our work. Once again, we sincerely appreciate the time and effort you spent reviewing our paper.

---

### Official Review · Reviewer_yvsu · 2024-07-09

**Soundness:** 2
**Presentation:** 3
**Contribution:** 2
**Rating:** 5
**Confidence:** 4

**Summary:**

This paper proposes Projected Diffusion Models (PDM) for constrained generative modeling. The key idea is to reframe the denoising process of diffusion models as a constrained optimization problem, iteratively projecting the generated samples onto a constraint set at every denoising step. The method is validated on several applications involving both convex and non-convex constraints, generally outperforming both conditional models and models that only project onto the constraint set after the last denoising step. The authors provide feasibility guarantees for convex constraints and optimality guarantees for convex constraints and likelihoods.

**Strengths:**

* The paper is well-written and structured. The authors clearly describe their method, the motivation behind it, and the experimental settings it is evaluated in.
* The method is straightforward to implement and compatible with pre-trained diffusion models, only requiring access to a projection operator at inference time.
* The experiments cover a wide range of applications and constraint types, including both convex and non-convex constraints, as well as in- and out-of-distribution scenarios.

**Weaknesses:**

* The paper heavily relies on Fréchet Inception Distance (FID) scores for the empirical comparison of models in Sections 5.1, 5.2, and 5.4. While FID is a standard metric for evaluating generative models on ImageNet-like images, it is unclear how meaningful it is in the specialized domains explored in this work (e.g., material microstructures, human poses, physics-informed simulations).
* The paper presents PDM as a novel optimization technique that "recasts traditional denoising strategies as a constrained optimization problem". However, if I understand it correctly, it is an application of commonly employed post-processing projections (e.g. the references provided in the manuscript) to the Langevin MCMC sampling scheme of Song and Earmon, NeurIPS 2019. It would be helpful if the authors could clarify this and outline any additional novelty/contributions.

**Questions:**

* The implementational details in Appendix F state that all experiments were carried out with $T=10$ diffusion time steps. This is a very small value, compared to the hundreds or thousands of time steps that are used in standard image diffusion papers. Is there a reason for this?
* How many samples were used to compute the quantitative performance metrics reported in Section 5? Would you expect a change in relative performance when using the same inference-time compute budget, i.e., generating 50% more conditional samples in the Constrained Materials application, since the conditional model is ~50% faster?
* Figure 2 visualizes the constraint satisfaction rate of a conditional model as a function of the relative error tolerance, given in percentage points. What does an error tolerance of 100% correspond to?

**Limitations:**

The main methodological limitation of the proposed technique is its increased computational cost. The authors adequately address this limitation and suggest different approaches to overcome it. As outlined above, I believe that the main limitation of the experimental evaluation presented in the manuscript is its reliance on the FID metric.

---

> ### Author Rebuttal · Authors · 2024-08-03
>
> *We will include abridged versions of our responses in this rebuttal window, but we ask that the reviewer refers to our complete answers in the comments.*
>
> Thank you for your time and efforts in providing feedback on our paper. We appreciate your acknowledgment of the diversity of our applications and constraint sets, as we consider this to be a significant component of our empirical validation of the proposed method. Let us emphasize that PDM reports state-of-the-art results in these diverse domains, outperforming baselines [5,11] in real-world microstructure synthesis when evaluated using FID, heuristic-based analysis, and constraint adherence, providing the first method we are aware of to _report zero-violations in human motion synthesis_, surpassing the current state-of-the-art in diffusion-based trajectory optimization reporting perfect feasibility and identical path length, and demonstrating applicability to video generation with complex, ODE-based constraints. Not only the range of domains we evaluate our method on exhibits its robustness across various constraint sets and data settings, but we also provide solid theoretical arguments for these behaviors. We hope you would agree that this demonstrates the generalizability of our framework and provides a compelling reason for why our paper should be positively considered.
>
> >**W1: The paper heavily relies on Fréchet Inception Distance (FID) scores for the empirical comparison...**
>
> First, we would like to point out, as the reviewer has acknowledged, that FID is a standard metric for evaluating generative models, and evaluation using this metric is _more than reasonable_. However, we believe the reviewer may have missed that **the paper indeed already reports several additional, domain specific metrics to supplement these results**. For Sections 5.1, 5.2, and 5.3 we include additional metrics (please see Section E.1, Section E.2, and Figure 6). Additionally, we will highlight that the baselines in Sections 5.2 and 5.4 use FID as their primary metric for evaluation ([29,30] and [26]).
>
> >**W2: The paper presents PDM as a novel optimization technique that "recasts traditional denoising strategies as a constrained optimization problem"... it is an application of commonly employed post-processing projections...**
>
> Note that, we have indeed acknoledged that some post-processing steps have been proposed previously, and indeed we compare aginst such methods. However, these methods post-process **after** the sampling process, which is a key difference, in light of our theoretical analysis. As discussed in Section 6, our approach is based on the insight that the cost of projection increases with the number of unconstrained steps (see also the illustration in Section E.3). Crucially, we have shown that post-processing approaches produce samples of much lower quality than those produced by PDM, improving material synthesis FID scores by over 30%, feasibility rates of tarjectory optimization by 90%, and increasing the quality of physics-based video generaiton by a factor of two. We argue theoretically, and empirically demonstrate, that a single post-processing projection leads to a significant divergence from the distribution in all the settings we examined. **This is a key novel contribution of this paper**.
>
> >**Q2: Would you expect a change in relative performance when using the same inference-time compute budget, i.e., generating 50% more conditional samples in the Constrained Materials application, since the conditional model is ~50% faster?**
>
> For the example that was referenced, as one may extrapolate from Figure 2, while the conditional model may generate 3000 samples within the time span it took for PDM to generate 2000 samples, if the error tolerance is less than ~35% (which is an absurdly high margin in this setting!!) then PDM generates many more feasible samples than the conditional model within the same compute budget. This discrepency is further emphasized when the tolerance is within a more reasonable margin, such as 5% where **PDM generates nearly seven times as many feasible samples within the same compute budget.** Hence, when constraints are integral to the outputs, PDM does, in fact, outperform conditional models in relative speed as well!
>
> >**Q3: Figure 2 visualizes the constraint satisfaction rate... what does an error tolerance of 100% correspond to?**
>
> When representing the porosity levels, we provide a percentage of pixels that should be below a provided threshold, as dark regions of the image represent damaged regions of the microstructure. For example, if an image had 50% porosity and 40% porosity was specified, the error tolerance that would make this feasible is 10%. Notice that our proposed method **guarantees** constraint adherence here, which is key for the scientific application tested.
>
> >**L1: The main methodological limitation... is its increased computational cost... the main limitation of the experimental evaluation presented in the manuscript is its reliance on the FID metric.**
>
> First, we would agree with the reviewer point about increased computational cost. This is an inherent byproduct of constraining any optimization problem, especially when providing guarantees upon the adherence to the constraint set. However, we would point you to our response to Question 2 for more context on this overhead. Additionally, we will note that in many settings, including those studied in this paper, *constraint-agnostic model cannot be used due to the necessity of adhering to the constraint set.*
>
> Second, we will point the reviewer to our response to Weakness 1. We hope that the reviewer will take the opportunity to revisit the additional metrics that we bring attention to here and also to examine the evaluation criteria used by the referenced baselines. As this is the primary justification provided for the score that the reviewer provided, we would ask you to consider raising your score with this provided context.

---

> ### Author Response · Authors · 2024-08-03
>
> >**Weakness 1: The paper heavily relies on Fréchet Inception Distance (FID) scores for the empirical comparison...**
>
> First, we would like to point out, as the reviewer has acknowledged, that FID is a standard metric for evaluating generative models, and evaluation using this metric is _more than reasonable_. However, we believe the reviewer may have missed the additional metrics we use which are tailored to specific domains.
> - For Section 5.1, we use **the same heuristic-based metrics used by Choi et al. [5]**; we report these in Section E.1, finding the realism of PDM's generations surpass the baselines using this evaluation as well.
> - Additional metrics for Section 5.2 are reported in Section E.2, although we will highlight that the compared baselines report FID as their primary metric for generation quality [29,30].
> - Section 5.3 uses **domain specific metrics from the state-of-the-art baseline [3]**.
> - Finally, we highlight that the primary metrics used by the baseline in Section 5.4 [26] is a variation of FID score, making this an appropriate point of comparison.
>
> Thus, we believe our use of FID score is indeed appropriate for the settings explored, especially as **the paper indeed already reports several additional metrics to supplement these results**.
>
> >**Weakness 2: The paper presents PDM as a novel optimization technique that "recasts traditional denoising strategies as a constrained optimization problem". However, if I understand it correctly, it is an application of commonly employed post-processing projections (e.g. the references provided in the manuscript) to the Langevin MCMC sampling scheme of Song and Earmon, NeurIPS 2019. It would be helpful if the authors could clarify this and outline any additional novelty/contributions.**
>
> Thank you for this question. As already discussed in our related work section, the novelty of this paper arises from (1) framing the reverse diffusion process as a constrained optimization problem, (2) formulating and implementing **general** projections for constraints and physical principle with important relevance for scientific and engineering applications, and (3) proposing a novel theoretical analysis to show that constraint adherance is not only feasbile, but guarantees can also be attained in many important classes of constraints with significant application relevance.
>
> Note that, we have indeed acknoledged that some post-processing steps have been proposed previously, and indeed we compare aginst such methods. However, these methods post-process **after** the sampling process, which is a key difference, in light of our theoretical analysis. As discussed in Section 6, our approach is based on the insight that the cost of projection increases with the number of unconstrained steps (see also the illustration in Section E.3). Theorem 6.2 supports this by showing that projection cost is lower when the sample starts from the feasible set, leading to better convergence properties. We explain that this is because high projection cost results in significant divergence from the distribution in all settings we explored. To our knowledge, this is the first theoretical result on constraint adherence in diffusion models, marking a key contribution of our work.
>
> Crucially, we have shown that post-processing approaches produce samples of much lower quality than those produced by PDM, improving material synthesis FID scores by over 30%, feasibility rates of tarjectory optimization by 90%, and increasing the quality of physics-based video generaiton by a factor of two. We argue theoretically, and empirically demonstrate, that a single post-processing projection leads to a significant divergence from the distribution in all the settings we examined. **This is a key novel contribution of this paper**.
>
> Additionally, similar to the theory behind projected gradient descent, using projections throughout the reverse process is advantageous as we strive to reach the constrained minimum. Once again, this is a key novel contribution of our work.
> Projections help guide the optimization process toward a region of the distribution that satisfies the constraints. Without regular projections, the optimization path may venture far outside the feasible space, potentially resulting in poor convergence properties, as we demonstrate in several experiments in the paper. In many ways projections guide the diffusion process of PDM in a similar spirit to the effect of conditioning *Cond* models, with the exception that this guidance imposes hard constraints on the generation process which we have shown are vastly more reliable than state-of-the-art conditioning techniques.
>
> (1/3)

---

> ### Author Response · Authors · 2024-08-03
>
> >**Q1: The implementational details in Appendix F state that all experiments were carried out with $T = 10$ diffusion time steps...**
>
> This is actually *standard for the score-based models we used*. Specifically, we would like to point the reviewer to Section 5 of [24] where they explain in the "Setup" paragraph that they set this value to 10 (note that these authors use $L = 10$, which is equivalent in their notation).
>
> >**Question 2: How many samples were used...? Would you expect a change in relative performance when using the same inference-time compute budget, i.e., generating 50% more conditional samples in the Constrained Materials application, since the conditional model is ~50% faster?**
>
> For the computed metrics, we used 2000 samples from each model for FID metrics and additional metrics provided in Section E. You bring up an interesting point as to the relative performance, which we assume to refer to the number of generated feasible samples, within a given given compute budget. For the example that was referenced, as one may extrapolate from Figure 2, while the conditional model may generate 3000 samples within the time span it took for PDM to generate 2000 samples, if the error tolerance is less than ~35% (which is an absurdly high margin in this setting!!) then PDM generates many more feasible samples than the conditional model within the same compute budget. This discrepency is further emphasized when the tolerance is within a more reasonable margin, such as 5% where **PDM generates nearly seven times as many feasible samples within the same compute budget.** Hence, when constraints are integral to the outputs, PDM does, in fact, outperform conditional models in relative speed as well!
>
> >**Q3: Figure 2 visualizes the constraint satisfaction rate... what does an error tolerance of 100% correspond to?**
>
> When representing the porosity levels, we provide a percentage of pixels that should be below a provided threshold, as dark regions of the image represent damaged regions of the microstructure. Note that in our generated data higher porosity levels have more dark regions than lower porosities. In Figure 2, the x-scale represents the deviation between the generated data's porosity and the specified porosity. For example, if an image had 50% porosity and 40% porosity was specified, the error tolerance that would make this feasible is 10%. A reasonable error tolerance would likely be below 10%, although for precision applications this would still be far be too high. Notice that our proposed method **guarantees** constraint adherance here, which is key for the scientific application tested.
>
> *A note on error tolerance in Figure 2:* We have been working directly with a material scientist collaborator in this  domain, and for their applications it is _necessary_ to generate results which report zero-violations, making wide error tolerances inviable for their work. This was a key motivation for our development of PDM. We would like to emphasize that generation of microstructures for energetic materials is a critical real-world problem in material science, presenting unique challenges such as data scarcity and the need to satisfy out-of-distribution constraints. This experiment has significant practical implications for the creation of new material structures, and our results are undergoing testing in laboratory settings. This speaks about the significance of our method.
>
> >**L1: The main methodological limitation... is its increased computational cost... the main limitation of the experimental evaluation presented in the manuscript is its reliance on the FID metric.**
>
> First, we would agree with the reviewer point about increased computational cost. This is an inherent byproduct of constraining any optimization problem, especially when providing guarantees upon the adherence to the constraint set. However, we would point you to our response to Question 2 for more context on this overhead. Additionally, we will note that in many settings, including those studied in this paper, *constraint-agnostic model cannot be used due to the necessity of adhering to the constraint set.* Our method is specifically tailored to scientific and engineering applications and the associated challenges including settings with low data, the absence of meaningful conditioning values, out-of-distribution generation, and particularly when exact constraint satisfaction in integral to the data quality. In such settings, conditional models often cannot be used.
>
> Second, we will point the reviewer to our response to Weakness 1. We hope that the reviewer will take the opportunity to revisit the additional metrics that we bring attention to here and also to examine the evaluation criteria used by the referenced baselines. As this is the primary justification provided for the score that the reviewer provided, we would ask you to consider raising your score with this provided context.
>
> (2/3)

---

> ### Author Response · Authors · 2024-08-05
>
> ---
> Once again, we would like to express our thanks for your review. We have addressed each question and concern presented in your review. We believe our thorough responses provide the necessary clarifications to justify the significance of our work.
> With these points in mind, we would greatly appreciate if you would re-evaluate your score to reflect the merit and significance of our contributions. If there are any additional specific reasons for the current assessment, we would appreciate further clarification and would be happy to discuss further. Thank you for your consideration!
>
> (3/3)

---

> ### Author Response · Authors · 2024-08-10
>
> We appreciate Reviewer yvsu’s detailed assessment and are grateful for the recognition of our _paper’s methodological clarity_ and _comprehensive experimental scope_.
> In our main rebuttal, we addressed the following key points:
> 1. **Use of FID Scores**: Our evaluation does not only consider FID scores (which is the standard metric for evaluating diffusion) but also included results on constraint satisfiability at various fidelity levels, and provided by additional domain-specific metrics reported in our experiments (see full response for details).
> 2. **Optimization Technique**: The distinctiveness of our method from traditional post-processing projections relies on the integration of projections directly into the Langevin dynamics, which is a novel contribution to the field. This approach not only improves sample quality but also ensures constraint adherence throughout the diffusion process. The theoretical analysis is also a important novel contribution of this work which justifies our modeling choice and provides guarantees for constraint adherence for important constraint classes.
> 3. **Computational Efficiency**: We addressed concerns about the use of a reduced number of diffusion steps (T=10), explaining that this setting is also what is adopted by previous score-based diffusion models and aligns with our model’s efficiency and sufficiency for achieving high-quality results. We also discussed how our method maintains efficiency compared to baselines, particularly when generating feasible samples within a given compute budget, showing that our model is, in effect, much faster in such settings.
>
> Are there any additional concerns that we could address? We are ready to provide further insights to assist in your evaluation and to enhance the understanding of our findings.

---

> > ### Author Response · Authors · 2024-08-11
> >
> > As the discussion period is nearing its end, we wanted to ask if there are any follow-up points we can clarify. Please also note our summary in the previous comment. Many thanks!

---

> ### Author Response · Authors · 2024-08-13
>
> Again, we would like to thank you for your assessment and are grateful for the recognition of our paper’s *methodological clarity* and comprehensive *experimental scope*. It has come to our attention that you have updated your score to a 5. As we have not had the opportunity to engage with you during this discussion period, we would like inquire as to what remaining concerns you have preventing you from advocating for strong acceptance? We believe we have addressed all the questions that were in your original review and would welcome the opportunity to discuss any outstanding doubts.

---

> > ### Comment · Reviewer_yvsu · 2024-08-13
> >
> > My apologies, there was an issue with the visibility settings of my original response. Here it is, including a discussion of my remaining concerns:
> >
> > I would like to thank the authors for the detailed response.
> >
> > ---
> >
> > **Re W1:** The paper heavily relies on Fréchet Inception Distance (FID) scores for the empirical comparison...
> >
> > > First, we would like to point out, as the reviewer has acknowledged, that FID is a standard metric for evaluating generative models, and evaluation using this metric is more than reasonable.
> >
> > Perhaps there is a misunderstanding. My concern was that FID is a standard metric for evaluating generative models **trained on natural image datasets**, since it relies on the last layer of an Inception v3 model that was trained on ImageNet. It is still unclear to me why it would be "more than reasonable" to expect these representations to facilitate meaningful performance comparisons when applied to material microstructures, 3D human poses, etc. since these application domains are strongly out-of-distribution with respect to the ImageNet data the Inception model was trained on.
> >
> > That being said, I appreciate the detailed clarifications and the pointers to alternative performance metrics. As far as I could tell, most of them are constraint satisfaction metrics (i.e. the *success percentage* in Figure 6 and the *penetrate* and *float* distances in Table 1), rather than sample quality metrics. However, the heuristics in Figure 13 do suggest that PDM matches the properties of the ground truth data better than conditional models on two of the three presented metrics. In light of this, I will raise my score by 1, but as outlined above my main concern regarding the applicability of FID to the settings in Sections 5.1, 5.2 and 5.4 remains.

---

> > > ### Author Response · Authors · 2024-08-13
> > >
> > > Thank you for clarifying where you stand with respect to our rebuttal. As demonstrated by the heuristic-based metrics in Figure 13, we have been intentional in selecting domain specific metrics for the experiments we have conducted. It appress that, as you have acknowledged the inclusion of these metrics for Section 5.1, your only remaining concern is the use of FID score in Sections 5.2 and 5.4.
> > >
> > > As we pointed out in our original response, the use of FID scores for these experiments *comes directly from the literature/baselines with which we compare*. FID score remains the predominant metric used for evaluation of the HumanML3D dataset used in Section 5.2. This is not a metric we just decided to use, but a metric that is necessary to compare to existing literature. While you may make the claim that this is not an appropriate metric, we would argue that this is the most appropriate metric as it is the only one that allows us to compare to existing work. **When working on these problems that the community is adopting as benchmarks it seems most appropriate to use the metrics that have already been selected.**
> > >
> > > A similar case can is made for Section 5.4. As this evaluation metric is used throughout the diffusion model literature, **and particularly in the literature we are comparing to,** we’d ask to please not punish us for something as accepted and recognized in the community.

---

### Official Review · Reviewer_e4oh · 2024-07-10

**Soundness:** 2
**Presentation:** 2
**Contribution:** 2
**Rating:** 4
**Confidence:** 3

**Summary:**

This paper proposed Projected Diffusion Models (PDM) inspired by stochastic gradient Langevin dynamics, to generate samples that satisfy given arbitrary constraints and remain within the specified regions. The authors claimed that the proposed algorithm is compatible across various applications, including satisfying morphometric properties when synthesizing materials, physics-informed motion generation, constrained path planning and human motion generation, and provided theoretical analysis to guarantee the generated samples reside in the constrained regions.

**Strengths:**

1. The approach this proposed achieves zero-violation to constraints during sampling process.
2. Compared against several existing approaches in the various experiments.

**Weaknesses:**

1. In limitation discussion, computational overhead is mentioned. How's the time complexity of PDM compared to ``Cond``, ``Cond+`` and ``Post+``?
2. Not quite sure if I really understand how ``Cond+`` is different from PDM in general except the Langevin dynamics part from the descriptions but perceive the differences between them and if I am correct, ``Cond`` is in classifier-free guidance regime, ``Cond+`` is in classifier guidance regime and ``Post+`` only applies projection once at the end of sampling process. It would be better to describe them using mathematical expressions.

Not a serious problem, but for the writing style, I found it a bit hard to follow when cited papers stays in the middle of the sentence, such as,  ``, emulating the post-processing approaches of [10, 21] in various domains presented next.`` from line 190-191, but I do not really know what 10 and 21 are about and what authors point to unless I check the reference list. It also looks a bit weird to me: ``The implementation of this model is as described by Ho and Salimans.``, where no number follows after the citation.

**Questions:**

Following the weaknesses above,

- For Equation (3b), it requires the whole trajectory in the reverse process satisfying the constraints, while $x_t$s' while $t$ is large are basically noise. Is this optimization setup reasonable?
- What's the time complexity to find the nearest feasible point while doing the projection? Will take it too long time? How do you choose $M$ in the algorithm 1 to control the number of Langevin Dynamics being applied?
- While doing projections, from my understanding, the projected feasible point should be on the boundary of the constraints. Does that mean the final samples will be all projected on the exact boundary then?

To summarize this question, even though PDM has better metrics than others, I still doubt the sampling distribution is not aligned with the data distribution since the samples might be all on the constraint boundaries.

- When evaluating out-of-distribution samples, is the constraint satisfaction rate the only metric?

**Limitations:**

Limitations and positive societal impact are discusses in the paper, but negative societal is not mentioned.

---

> ### Author Rebuttal · Authors · 2024-08-03
>
> *We will include abridged versions of our responses in this rebuttal window, but we ask that the reviewer refers to our complete answers in the comments.*
>
> Thank you for your valuable feedback. Before addressing your specific questions, let us emphasize the significant contribution provided by our work: Our proposed method provides constraint imposition with _formal guarantees_ for several important classes and for the first time in the general and complex form posed! All prior work has either been unable to cope with the complex constraints that are crucial for the scientific and engineering application of interest (which include ODEs, nonconvex constraint sets, and real-world, scientific applications), relied on costly post-processing methods that we have shown _fail to produce meaningful samples in the diverse and real-world set_ of studied domains, and have failed to supply formal guarantees of constraint satisfaction as provided by our work.
>
> >**W1: How's the time complexity of PDM compared to Cond, Cond+ and Post+?**
>
> We report the runtime difference between PDM and the other baselines in Section F. Additional details on time complexity of our projections are included in the comments below. Also, we will refer you to our response to Q2 by Reviewer yvsu, as this is closely related to your question of additional overhead (*as our PDM is much faster than conditional models when constraint adherence is necessary to generate viable samples!*).
>
> >**W2: Unclear how Cond+ is different from PDM....**
>
> To clarify, *Cond* is a classifier-free guidance. The conditional model uses the guidance scheme provided at the end of page 2 (line 83). *Cond+* and *Post+* are equivalent to current post-processing methods, and we would like to emphasize the significant performance gap between these methods and PDM. We appreciate your suggestion of including a more clear formalization of these methods in the paper and will do so in our final version of the manuscript. *More details provided in the comments.*
>
> >**Q1: For Equation (3b), it requires the whole trajectory in the reverse process satisfying the constraints, while $x_t$s' while $t$ is large are basically noise. Is this optimization setup reasonable?**
>
> Theoretically, as discussed in Section 6, this approach is based on the insight that the cost of projection increases with the number of unconstrained steps. Theorem 6.2 supports this. Importantly, this theorem shows that the projection cost is lower when the sample starts from the feasible set, leading to better convergence properties. This is further demonstrated empirically, as methods imposing constraints only at the final step perform significantly worse than PDM, especially in non-convex settings. Importantly, "performing worse" here refers *not* to constraint adherence, but to the ability to generate images from the original data distribution (i.e., high-fidelity outputs) while satisfying the imposed constraints. Indeed, we showed how projecting only in the last step results in a significant divergence from the distribution in all settings we explored. Additionally, paralleling the theory behind projected gradient descent, as we aim to converge to the constrained minimum, there is a clear benefit in using projections to guide the optimization process toward a region where the minimum satisfying the constraints can be found. Practically, we've observed that unconstrained optimization steps can deviate significantly from the feasible domain, making it difficult to converge to a feasible sample. This is effectively illustrated in Figure 1 in the paper, where our proposed guidance scheme, PDM, navigates the constraint-defined landscape to ensure convergence to a feasible sub-distribution.
>
>
> >**Question 3: ... I still doubt the sampling distribution is not aligned with the data distribution since the samples might be all on the constraint boundaries.**
>
> The projection will provide a point on the boundary of the constraints *if the original sample was infeasible.* In effect, this is why *Post+* and *Cond+* report such high FID scores, as more likely than not (for example, see Figures 2 and 8) prior to post-processing these samples are infeasible. In contrast, PDM samples converge to feasible subdistributions. For instance, notice that in Figure 1 later timesteps result in no violations of the constraints. This implies that unless subsequent gradient steps were consistently along the constraint boundaries (for all the samples visualized), *the samples output did not fall on the constraint boundaries and instead were from unique points within the feasible subdistribution.* PDM samples are "moved" to a subset of the distribution which is still optimal (maximizes the density function) but is also feasible. The projections solely enforce that the generated samples are taken from this region of the distribution.
>
> Furthermore, FID scores are widely used in image generation tasks because of how well they benchmark the similarity between distributions. This metric is particularly useful because **it reliably asesses both sample quality and diversity;** models which lack sample diversity perform very poorly when reporting FID scores. It is _widely accepted that this is the most appropriate method for assessing how well the sampling distributions and data distributions align_ and we show how well our proposed method performs in FID scores, while also satisfying the imposed constraints.
>
> >**Q4: When evaluating out-of-distribution samples, is the constraint satisfaction rate the only metric?**
>
> In this setting, the FID scores of the out-of-distribution generations did not change from the scores reported for the in-distribution generations. Hence, we do not report these separately.
>
> ---
> If there are any additional reasons for the current assessment, we would appreciate further justification to address any remaining concerns. Thank you!

---

> ### Author Response · Authors · 2024-08-03
>
> >**Weakness 1: In limitation discussion, computational overhead is mentioned. How's the time complexity of PDM compared to Cond, Cond+ and Post+?**
>
> We report the runtime difference between PDM and the other baselines in Section F. We would be happy to provide details on the time complexity of our projection methods:
> - **Constrained Materials:** This is a knapsack constraint. While in general knapsack problems are NP-complete, the version adopted (which uses integers as weight) is known to be weakly NP-complete and admits a fully polynomial approximation. It is solved efficiently in $O(nm)$ where $n$ is the number of pixels and $m$ is the number of values to compute for the dynamic program.
> - **3D Human Motion:** This is a scaling constraint that runs in $O(n)$ time, where $n$ is the length of the internal representation.
> - **Constrained Trajectories:** The interior point method used by the nonconvex solver in our implementation [27] has a time complexity of $O(n^{3.5})$ where $n$ is the number of variables in the Quadratic Program. In this case, there are 128.
> - **Physics-informed Motion:** This projection runs in $O(n+m)$ where the size of each image is $n$ by $m$.
>
> As we note in Section F, these projection operations have not been optimized for runtime, and these time complexities represent an upper bound. Additionally, we will refer you to our response to Question 2 by Reviewer yvsu, as this is closely related to your question of additional overhead (*as our PDM is much faster than conditional models when constraint adherence is necessary to generate viable samples!*).
>
> >**Weakness 2: Unclear how Cond+ is different from PDM in general except the Langevin dynamics part from the descriptions but perceive the differences between them and if I am correct, Cond is in classifier-free guidance regime, Cond+ is in classifier guidance regime and Post+ only applies projection once at the end of sampling process. It would be better to describe them using mathematical expressions.**
>
> To clarify,
> - *Cond:* is a classifier-free guidance. The conditional model uses the guidance scheme provided at the end of page 2 (line 83).
> - *Cond+:* This model is identical to *Cond*, but the final output $x_1$ is projected using our projection operator $\mathcal{P}_C$. Thus, this is a baseline introduced by this work and inspired by generative models using post-processing steps.
> - *Post+:* This is an unconditioned score-based model of identical architecture to PDM. Instead of constraining the entire generation as with PDM, we project only on the final output $x_1$.
>
> *Cond+* and *Post+* are equivalent to current post-processing methods, and we would like to emphasize the significant performance gap between these methods and PDM. We appreciate your suggestion of including a more clear formalization of these methods in the paper and will do so in our final version of the manuscript.
>
> (1/3)

---

> ### Author Response · Authors · 2024-08-03
>
> >**Question 1: For Equation (3b), it requires the whole trajectory in the reverse process satisfying the constraints, while $x_t$s' while $t$ is large are basically noise. Is this optimization setup reasonable?**
>
> Our decision to enforce constraints at every step of the denoising process is driven by theoretical insights as well as practical observations.
>
> Theoretically, as discussed in Section 6, this approach is based on the insight that the cost of projection increases with the number of unconstrained steps (also see illustration in Section E.3). Theorem 6.2 supports this. importantly, this theorem shows that the projection cost is lower when the sample starts from the feasible set, leading to better convergence properties. This is further demonstrated empirically, as methods imposing constraints only at the final step perform significantly worse than PDM, especially in non-convex settings. Importantly, "performing worse" here refers *not* to constraint adherence, but to the ability to generate images from the original data distribution (i.e., high-fidelity outputs) while satisfying the imposed constraints. Indeed, we showed how projecting only in the last step results in a significant divergence from the distribution in all settings we explored. Additionally, paralleling the theory behind projected gradient descent, as we aim to converge to the constrained minimum, there is a clear benefit in using projections to guide the optimization process toward a region where the minimum satisfying the constraints can be found. Without regular projections, the optimization path may explore regions far outside the feasible space, potentially leading to poor convergence properties.
> We also notice that this is the first theoretical result on constraint adherence in diffusion models, to the best of our knowledge, and is indeed a key contribution of our work.
>
> Practically, we've observed that unconstrained optimization steps can deviate significantly from the feasible domain, making it difficult to converge to a feasible sample. This is effectively illustrated in Figure 1 in the paper, where our proposed guidance scheme, PDM, navigates the constraint-defined landscape to ensure convergence to a feasible sub-distribution.
>
> Notice also that continuous projection throughout the sampling process is crucial for converging to feasible solutions in non-convex settings, as shown in Section 5.3. In these experiments, projecting throughout the sampling process allows our method to converge to feasible solutions consistently (i.e., **we never produced an unsatisfiable trajectory**!) with a single sample. In contrast, as shown in Figure 6, the *Cond+* method, a state of the art method introduced to solve this specific problem in [21] which imposes "constraints" (a post-processing step) only at the final step, was never able to correct the infeasible samples, with the solver repeatedly reporting local infeasibility. This alone should be considered as a substantial improvement over the state-of-the-art.
>
> >**Question 2: What's the time complexity to find the nearest feasible point while doing the projection? Will take it too long time? How do you choose $M$ in the algorithm 1 to control the number of Langevin Dynamics being applied?**
>
> For the first part of this question, please refer to our response for Weakness 1.
>
> To answer the latter part, we use the value of $M$ used by Song et al. [24] when this architecture was proposed, finding that the samples effectively converge using this value in our experiments. Analysis of an optimal value for $M$ could be conducted but is out of the scope of the question investigated in this work, which is on demonstrating a new constraint and physical principle adherence in generative models.
>
> (2/3)

---

> ### Author Response · Authors · 2024-08-05
>
> >**Question 3: While doing projections, from my understanding, the projected feasible point should be on the boundary of the constraints. Does that mean the final samples will be all projected on the exact boundary then? To summarize this question, even though PDM has better metrics than others, I still doubt the sampling distribution is not aligned with the data distribution since the samples might be all on the constraint boundaries.**
>
> The projection will provide a point on the boundary of the constraints *if the original sample was infeasible.* In effect, this is why *Post+* and *Cond+* report such high FID scores, as more likely than not (for example, see Figures 2 and 8) prior to post-processing these samples are infeasible. In contrast, PDM samples converge to feasible subdistributions. For instance, notice that in Figure 1 later timesteps result in no violations of the constraints. This implies that unless subsequent gradient steps were consistently along the constraint boundaries (for all the samples visualized), *the samples output did not fall on the constraint boundaries and instead were from unique points within the feasible subdistribution.* The effect of guiding the sampling with projections is that PDM samples are "moved" to a subset of the distribution which is still optimal (maximizes the density function) but is also feasible. The projections solely enforce that the generated samples are taken from this region of the distribution.
>
> Furthermore, FID scores are widely used in image generation tasks because of how well they benchmark the similarity between distributions. This metric is particularly useful because *it reliably asesses both sample quality and diversity;* models which lack sample diversity perform very poorly when reporting FID scores. It is _widely accepted that this is the most appropriate method for assessing how well the sampling distributions and data distributions align_ and we show how well our proposed method performs in FID scores, while also satisfying the imposed constraints. We believe this is a huge strength of the proposed approach and hope you see its significance given the many scientific and engineering domains in which this can be adopted, as we demonstrate in our experiments.
>
> >**Question 4: When evaluating out-of-distribution samples, is the constraint satisfaction rate the only metric?**
>
> When answering this, we are presuming you are referring specifically to Section 5.4, although please correct us if that is not the case. We also report FID scores here; in this setting, the FID scores of the out-of-distribution generations did not change from the scores reported for the in-distribution generations. Hence, we do not report these separately, but they are applicable to both settings.
>
> Additionally, we'll note that similarly various out-of-distribution constraints are added in Section 5.3 (the red obstacles in Figure 5 which were not present in the training data). Our method is equally robust in these settings, with consistent results across the studied topographies.
>
> ---
> We believe that our detailed responses provide the necessary clarifications to all your questions and reinforce the robustness and significance of our work. In light of these clarifications, we kindly request that you re-evaluate your score to reflect the merit and significance of our contributions. Our work makes substantial contribution for the application of generative processes for various engineering and scientific application settings requiring satisfaction of constraints and physical rules, as we demonstrate in the paper.
>
> If there are any additional, specific reasons for the current assessment, we would appreciate further justification to understand and address any remaining concerns. Thank you for your consideration!
>
> (3/3)

---

> ### Author Response · Authors · 2024-08-10
>
> We appreciate the feedback provided by Reviewer e4oh and are grateful that _our proposed method’s effectiveness and the theoretical underpinnings_ were recognized as strengths of our work.
> In our main rebuttal, we addressed the following key points:
> 1. **Methodological Differences and Formalizations**: We provided a detailed explanation to distinguish our method from the baseline proposed: Cond+, Cond, and Post+ techniques. These, together with additional mathematical expressions will also be reflected in the final version of our paper.
> 2. **Computational Overhead**: We compared the time complexity of our Projected Diffusion Models with other models and highlighted the efficiency of our approach, especially when constraint adherence is critical, which is the case in all setting studied. For instance, in Section 5.1, with the error tolerance is within 5%, our method generates nearly 7 times as many feasible samples within the same compute budget.
> 3. **Optimization Setup**: As also pointed out in the paper, we remarked why projections needs to be applied to the entire trajectory. Our response included a theoretical backing and empirical evidence demonstrating the effectiveness of our approach, particularly in avoiding divergences from the target distribution.
> 4. **Sample Diversity**: We clarified how our projections guide the sampling to a feasible subdistribution (Figure 1), addressing concerns as to samples falling on the constraint boundary. Furthermore, we have explained how this is captured in the FID score - as it is widely accepted that this is the most appropriate method for assessing how well the sampling distributions and data distributions align.
>
> Are there any additional concerns that we could address? We are ready to provide further insights to assist in your evaluation and to enhance the understanding of our findings.

---

> > ### Author Response · Authors · 2024-08-11
> >
> > As the discussion period is nearing its end, we wanted to ask if there are any follow-up points we can clarify. Please also note our summary in the previous comment. Many thanks!

---

> > > ### Comment · Reviewer_e4oh · 2024-08-12
> > >
> > > I thank the authors' responses; however, the explanations confirms my doubt and concerns that the extra projection of the sample into the constrained set after every Langevin dynamics is not rational from my perspective. Theoretically, the step size in the Langevin dynamics should be chosen very carefully such that the injected noise brings the randomness while not messing up the effect from the gradient step. in this case, there is no guarantee, and I also kindly disagree that the projection can still maintain the sample at its' noisy level and this will definitely mess up the sampling scheme, or the noise level should be correspondingly adjusted according to the projection but that seems infeasible. This can be shown from the some metrics that evaluates the sample quality and how well they aligned with the data distribution . For example, the 3D human motion experiment from section 5.2 has worse FID score compared to the baseline that the paper provided, which is 0.71 (PDM) versus 0.63 (baseline). This paper also mentioned [1], which couldn't guarantee the samples always falling into the constrained region, but has much lower and better FID score, 0.551 and 0.433 as what they reported in [1].
> > >
> > > I understand that every sample generated by PDM that this paper proposed satisfy the constraints, but in my point of view, we shouldn't sacrifice the sample quality to achieve the goal. More or less, the proof provided in the section 6 assumes the feasible region is convex, which is very strict and doesn't apply in many cases.
> > >
> > >
> > > [1] Ye Yuan, Jiaming Song, Umar Iqbal, Arash Vahdat, and Jan Kautz. Physdiff: Physics-guided
> > > 460 human motion diffusion model. In Proceedings of the IEEE/CVF International Conference on
> > > 461 Computer Vision, pages 16010–16021, 2023.

---

> ### Author Response · Authors · 2024-08-12
>
> Thank you for continuing to engage with us during this discussion phase. We would like to take the opportunity to respond to the points you have made.
> > [W]e shouldn’t sacrifice the sample quality to [satisfy the constraints.]
>
> While this may be the case for some applications - such as recreational image generation, the opposite is true in scientific and engineering applications that motivate this study. When considering the practicality of diffusion models to these domains, the ability to satisfy constraints is integral to the viability of the outputs! In such settings, high quality samples are desirable, making the FID score still relevant, **but feasible samples are necessary**. Thus, it is not sufficient to merely provides samples which report well on this single metric, making our method stand out. *Furthermore, if FID was the only metric that mattered in these settings, post-processing methods would never have been proposed in previous work, as empirically we have demonstrated that these provide much worse FID scores.* Yet these methods have still found an important place in existing discourse!
>
> Note that this paper deals with real scientific and engineering settings presenting unique challenges such as data scarcity and the need to satisfy out-of-distribution constraints. In many scientific applications, such as the material science application we study, data collection is extremely expensive (it means synthetizing new materials) and the data collected may not provide feasible samples. This is exactly the in our material science application (see Section 5.1). There, it is essential that exact morphomoteric properties are satisfied, but these settings were never observed nor measured before!
> We remind the reviewer here that these constraints are imposed by expert material scientist with whom we collaborate and whose work particularly motivated this paper. The settings explored require stringent constraint adherence for the laboratory testing (which our work is currently undergoing attesting for the broader impact of our method).
>
> Similarly, constraints are necessary to generate accurate simulations (Sections 5.2 and 5.4) and clearly integral to trajectory optimization. Samples which are not feasible provide no merit when exact constraint satisfaction must be considered. The ODE constrains enforced to capture the motion of free-falling object also allow us to generate objects falling on other planets, when such data is unlikely to be collected (see our simulations in Section 5.4).
>
> _Next, let us note a simple but important fact pertaining to constrained optimization in general_: when imposing constraints on an optimization procedure, inherently the feasibility space shrinks, and thus, of course, the cost objective typically is higher (in case of a minimization procedure) than that associated to an unconstrained one. However, when constraints are part of the optimization problem, solutions (or samples) which do not satisfy the constraints **cannot be considered “optimal”, or better solutions than feasible ones**. In our work, infeasible samples are not viable.
>
> We would also like to highlight that the “much lower and better FID score” that is referred to in the response cannot be directly compared as Yuan et al. uses an **external simulator to post-process their outputs**; here, the simulator takes as input a vector of points produced by the generative model and is crucial to produce realistic results. This  dramatically alters the diffusion model’s outputs! The baseline we provide is a more apt point of comparison with regard to the FID scores because it does not rely on external simulators, thus effectively compares the ability of generative models.
>
> Also note that in all experiments our FID scores are *very close or outperform the conditional model* (for instance Section 5.1 outperforms in terms of FID and heuristic-based metrics and Section 5.3 outperforms in the domain specific metrics). Furthermore, given that in many of these settings the training data is not necessarily feasible, sampling from a subdistribution of this data set will inherently result in FID score increases (as the diversity of the samples is limited by some margin for this conditional distribution).
> While it may make sense to still use a conditional model in settings where constraint adherence is not valued, we will reiterate that in many settings this necessary. Hence, we disagree with the “blanket statement” provided.
>
> (1/2)

---

> ### Author Response · Authors · 2024-08-12
>
> > [T]he proof provided in the section 6 assumes the feasible region is convex, which is very strict and doesn’t apply in many cases.
>
> From an optimization perspective, assuming convexity of the constraint set is more than reasonable. In fact, when providing convergence guarantees, this is almost universally assumed (as guaranteeing convergence of non-convex constraint sets is an unsolved problem).
> While the theoretical guarantees hold for convex settings, our experiments extend to non-convex constraints as well (see Section 5.3). In constraint optimization, this is reality that is widely accepted as also supported empirically by our results.
> Please notice that *inclusion of theoretical results is meant to strengthen the understanding of our work, and should not be considered as a weakness.*
>
> ---
> We hope our responses clarify your points. Are there any additional concerns that we could address? We are ready to provide further insights as needed.
>
> (2/2)

---

> > ### Comment · Reviewer_e4oh · 2024-08-13
> >
> > Thank you for your response forwards my concerns. I still believe the algorithm that directly projecting sample to the feasible region at every Langevin dynamics sampling step is the **flaw** it possesses, which can be reflected onto the **high FID score** from experiment 5.1. As what I mentioned from the last round of response, ``Theoretically, the step size in the Langevin dynamics should be chosen very carefully such that the injected noise brings the randomness while not messing up the effect from the gradient step. in this case, there is no guarantee, and I also kindly disagree that the projection can still maintain the sample at its' noisy level and this will definitely mess up the sampling scheme, or the noise level should be correspondingly adjusted according to the projection but that seems infeasible`` is the reason why **the direct projection will make the algorithm deficient**, but not all projection ideas would not work. For example, [1] also has the idea that bounces the samples back to the feasible regions through the whole reverse sampling process but it is with reasonable assumptions and justifies why this would not mess up the sampling scheme as well, which makes more sense. [1] also pointed out, in its introduction section, ``Although thresholding avoids failure, it is theoretically unprincipled because it leads to a mismatch between the training and generative processes.``, which is also the information that I tried to convey for the whole time.
> >
> > I firmly believe that the samples satisfying constraints are necessary, but if they are not generated in the way that follows the principle of how diffusion models train and sample, and how the theory works behind, the samples are meaningless from my point of view.
> >
> > [1] Lou, Aaron, and Stefano Ermon. "Reflected diffusion models." International Conference on Machine Learning. PMLR, 2023.

---

> > > ### Author Response · Authors · 2024-08-13
> > >
> > > Thank you for your continued discussion with us.
> > >
> > > Firstly, please note that we report very strong empirical results in our paper, demonstrating state-of-the-art in Section 5.1 and 5.3 *(we believe you referencing “high FID score from experiment 5.1” may be a typo here)* and state-of-the-art for constraint satisfaction methods on Sections 5.2 and 5.4, dramatically out-performing post-processing techniques which have found an important place in the literature.
> > >
> > > Your concerns seem now to be more theoretically-based. First, we would encourage you to look at our discussion with Reviewer Jdpw, where we provide additional discussion for our formulation of the reverse process as an optimization problem. We have **proved convergence properties of our method in Section 6, under standard convexity assumptions**. However, let us go further - **[A] proves convergence for non-convex optimization using Langevin dynamics**, making our method broadly applicable, and again our strong results are empirical evidence of this.
> > >
> > > As we have shown that the reverse process is the minimization of $- \log q(x)$, our method adapts the reverse process from a simple variation of gradient descent (SGLD) to a variation of projected gradient descent. **Projected gradient descent is known to converge (even in non-convex settings)** [B].
> > >
> > > Our samples are not “meaningless”. We have provided rigorous theoretical justification (and an excess of empirical results), and provided evidence that this work could be a significant resource to adopt generative models in many engineering and scientific domains where physical principles and user-imposed constraint must be satisfied for the outputs to be considered as “valid”. We do hope that you agree with us on the significance and potential broader impact of these results, given the evidence provided.
> > >
> > > ---
> > > [A] Raginsky, Maxim, Alexander Rakhlin, and Matus Telgarsky. "Non-convex learning via stochastic gradient langevin dynamics: a nonasymptotic analysis." Conference on Learning Theory. PMLR, 2017.
> > >
> > > [B] Vu, Trung, Raviv Raich, and Xiao Fu. "On convergence of projected gradient descent for minimizing a large-scale quadratic over the unit sphere." 2019 IEEE 29th International Workshop on Machine Learning for Signal Processing (MLSP). IEEE, 2019.

---

### Official Review · Reviewer_Jdpw · 2024-07-12

**Soundness:** 2
**Presentation:** 3
**Contribution:** 3
**Rating:** 5
**Confidence:** 3

**Summary:**

This paper proposes a diffusion model that imposes constraints on the generated output. However, the constraints here are not abstract verbal instructions but rather formalizable constraints. The authors propose projected diffusion model sampling to perform constraint conditional log-likelihood maximization at each time step. They demonstrate the effectiveness of the proposed method on a variety of application problems.

**Strengths:**

* The paper demonstrates the effectiveness of the proposed method under various constraint conditions.
* Section F reports on the impact of projection on computational cost.
* The paper compares the proposed method to appropriate existing methods, such as post-processing correction.

**Weaknesses:**

1. Equation (3) is introduced without a clear explanation of the relationship between the reverse diffusion process of score-based models and maximizing the conditional density function.
2. The details of the projection algorithm used in each experiment are only described in words. Pseudocode would be helpful.

**Questions:**

1. Please provide a more detailed explanation of the similarity between the reverse diffusion process of score-based models and maximizing the conditional density function.
2. As mentioned in Reference [20], it is common to gradually decrease the learning rate and $\gamma$ to improve the convergence rate of the proximal algorithm. Have you tried decreasing $\gamma$ as $1/i$ or $1 / \sqrt i$ within the inner loop of projected diffusion model sampling?
3. Please provide the computational order of the projection algorithm used in each experiment.

**Limitations:**

Section 7 adequately describes the limitations.

---

> ### Author Rebuttal · Authors · 2024-08-03
>
> Thank you for your time and efforts in providing feedback on our paper. First, let us emphasize the significant contribution provided by our work: Our proposed method provides constraint imposition with _formal guarantees_ for several important classes and for the first time in the general and complex form posed! All prior work has either been unable to cope with the complex constraints that are crucial for the scientific and engineering application of interest (which include ODEs, nonconvex constraint sets, and real-world, scientific applications), relied on costly post-processing methods that we have shown _fail to produce meaningful samples in the diverse and real-world set_ of studied domains, and have failed to supply formal guarantees of constraint satisfaction. We appreciate your acknowledgment of our use of "formalizable constraints," as we view this as a key contribution that distinguishes our work from existing literature. Let us reiterate that the method we propose is able to handle _arbitrary constraint sets_, meaning that this can generalize to *any formal constraints that are posed.* We believe these to be compelling reasons as to why our paper should be considered!
>
> >**Weakness 1 and Question 1: Equation (3) is introduced without a clear explanation of the relationship between the reverse diffusion process of score-based models and maximizing the conditional density function.**
>
> The objective of the reverse diffusion process is to maximize the density function, and Equation 4 shows the _actual_ update steps that are learned by the score-based model: a noisy (SGLD) gradient ascent on the density function. The formalization of the objective provided in Equation (3a) is equivalent by construction to that of the reverse diffusion process as _this is the optimization procedure that is learned during the forward diffusion process_.
> The model has learned to estimate the gradients necessary to solve this optimization problem, and the update step in Equation 4, which is taken directly from Song et al. [24,25], converges to a solution (or sample) that maximizes the density function. **In short, Equation (3a) is how the reverse diffusion process would be directly formalized as an optimization problem, and Equation (3b) is our method's extension of this objective to a constrained optimization problem.** We hope this clarifies your doubt.
>
> >**Weakness 2: The details of the projection algorithm used in each experiment are only described in words. Pseudocode would be helpful.**
>
> We appreciate the interest in the lower-level implementation details. First, notice that the actual implementations for each of these projections **are indeed included in our submission**, and if you are interested we would encourage you to look at the provided code. Additionally, we would encourage you to refer to our response to _Reviewer p9zr Weakness 1_ where we provide the mathematical formalization of the projections.
>
> >**Question 2: As mentioned in Reference [20], it is common to gradually decrease the learning rate and $\gamma$ to improve the convergence rate of the proximal algorithm. Have you tried decreasing $\gamma$ as $1/i$ or $1/\sqrt{i}$ within the inner loop of projected diffusion model sampling?**
>
> Indeed, our implementation in Section D (Algorithm 2) does provide a dynamic adjustment of $\gamma$ which uses stochastic differential equations to create smoother transition kernels between timesteps [25]. We find that for our physics-informed motion experiments this produces better FID scores, but in most of our experiments, $\gamma$ has been adjusted at the outer loop level, thus while decreasing with time it does not provide as a smooth of a change as that explored in Section D, without producing meaningful alteration to the algorithm performance.
>
> >**Question 3: Please provide the computational order of the projection algorithm used in each experiment.**
>
> We note that additional details for each projection are also reported in Section C, as we mentioned in the main paper. We provide here additional details on the problems' complexity:
> - **Constrained Materials:** This is a knapsack constraint. While in general knapsack problems are NP-complete, the version adopted (which uses integers as weight) is known to be weakly NP-complete and admits a fully polynomial approximation. It is solved efficiently in $O(nm)$ where $n$ is the number of pixels and $m$ is the number of values to compute for the dynamic program.
> - **3D Human Motion:** This is a scaling constraint that runs in $O(n)$ time, where $n$ is the length of the internal representation.
> - **Constrained Trajectories:** The interior point method used by the nonconvex solver in our implementation [27] has a time complexity of $O(n^{3.5})$ where $n$ is the number of variables in the Quadratic Program. In this case, there are 128.
> - **Physics-informed Motion:** This projection runs in $O(n+m)$ where the size of each image is $n$ by $m$.
>
> We will be happy to add this additional information to the appendix in our final version.
>
> ---
> We appreciate the constructive feedback and have diligently addressed all the concerns raised. We believe that our detailed responses provide the necessary clarifications for the arguments presented in our paper. In light of our clarifications, we kindly request that you re-evaluate your score, particularly given that the only weaknesses identified were minor points that have now been fully addressed.
>
> We also noticed your current score for soundness = 1. This appears to be unjustified, especially considering the rigorous methodological approach and theoretical contributions presented. If the score was intended to be revised following our responses, we hope that the comprehensive explanations provided will justify a favorable assessment. If there are any lingering doubts or additional questions, we are more than willing to discuss them further! Thank you.

---

> > ### Comment · Reviewer_Jdpw · 2024-08-12
> >
> > Thank you for your detailed response.
> >
> > > The objective of the reverse diffusion process is to maximize the density function, and Equation 4 shows the actual update steps that are learned by the score-based model: a noisy (SGLD) gradient ascent on the density function.
> >
> > Please cite papers that state that the objective of the reverse diffusion process is density function maximization.
> > I cannot find that the score-based methods maximize the likelihood of the density function in [24,25].
> > I believe Song et al. [24,25] are not likely-based. By modeling the score function instead of the density function, we can sidestep the difficulty of intractable normalizing constants of the density function.
> >
> > My current score for soundness = 1 reflects the above concern about Equation (3).
> >
> > > Indeed, our implementation in Section D (Algorithm 2) ...
> >
> > Since the $\gamma$ in Algorithm 2 is not always decreasing, your answer to Question 2 is no, right?
> > I don't think it's a problem that the answer to this question doesn't significantly detract from the main contributions.
> >
> > >  Additionally, we would encourage you to refer to our response to Reviewer p9zr Weakness 1 where we provide the mathematical formalization of the projections.
> >
> > Thank you for providing the answer to Question 3 and the mathematical formalization of the projections to p9zr. It is very helpful to understand the overview of the projection algorithms, and I recommend including them in the main text.

---

> ### Author Response · Authors · 2024-08-10
>
> We sincerely appreciate Reviewer Jdpw’s thorough evaluation and are grateful that _our method’s effectiveness under various constraints_ and the _comprehensive coverage of computational cost impacts_ were recognized.
> In our main rebuttal, we addressed the following key points:
> 1. **Clarification on Equation (3)**: We provided a detailed explanation of how Equation (3) relates to maximizing the conditional density function through the reverse diffusion process. The subsequent equations and the update steps, explained in our rebuttal, demonstrate the rigorous theoretical underpinnings of our model.
> 2. **Details of Projection Algorithm**: We have provided references to the detailed mathematical formalization and the actual code in our submission, ensuring that the implementation details are accessible and transparent. We will also provide additional formalization in the paper, based on the response above.
> 3. **Dynamic Adjustment of Parameters**: We clarified our method’s approach to parameter adjustment (which indeed explores a dynamic adjustment of $\gamma$ in Appendix D). A discussion on how these adjustments affect the model’s performance across different experimental setups is also included.
>
> Are there any additional concerns that we could address? We are eager to provide further information to assist in your evaluation and to enhance the understanding of our findings.

---

> > ### Author Response · Authors · 2024-08-11
> >
> > As the discussion period is nearing its end, we wanted to ask if there are any follow-up points we can clarify. Please also note our summary in the previous comment. Many thanks!

---

> ### Author Response · Authors · 2024-08-12
>
> Thank you for your response and allowing us the opportunity to clarify further:
> > The objective of the reverse diffusion process is to maximize the density function
>
> First, let us clarify that the papers we cited [24,25] are to point out that the update step taken in our paper (before projecting) is identical to those in existing literature (for instance it is also Equation 4 in [24]). You will also notice that our Algorithm 1 directly employs this update step (line 6) as in [24]. Next, notice that immediately following the introduction of this update step in [24] it is explained that the sample converges to $p(x)$ by the repeated application of this update step under the regularity conditions in [A]. These conditions are introduced in Equation 2 of [A] and are expressly for the purpose of “convergence to a local maximum” by ensuring that the solution “will reach the high probability regions”.
>
> This can also be directly derived from SGLD. For instance, refer to the formalization of SGLD in [B] Equation 1.2:
>
> $$
> d\mathbf{X}(t) = -\nabla F_n (\mathbf{X}(t))dt + \sqrt{2\beta^{-1}}d\mathbf{B}(t)
> $$
>
> First, notice that this matches the update step in Equation 4 (of our paper and [24]). As explained in [B], this optimization procedure “concentrates around the global minimum of $F_n(x)$” and parallels the task of directly minimizing $F_n$. Adapted to our update step (as shown below), this becomes the minimization of $-\log q(x)$ or, in other words, the maximization of the density function.
>
> $$
> d\mathbf{X}(t) = \nabla \log q (\mathbf{X}(t))dt + \sqrt{2\gamma(t)}d\epsilon(t)
> $$
>
> Additionally, we will note that [C] provides a complete proof showing that SGLD converges to an "almost-minimizer" of the function, as the noise maintains some degree of stochasticity. Hence, from these works we can support our claim that this process is "akin to maximizing the density function".
>
> Importantly, we would like to remind the reviewer that formalizing the reverse process as an optimization problem is part of the novelty of our work. Hence, existing papers have not explicitly presented the reverse process as in our work. But *the presentation of this objective in our paper is posed to show how can one incorporate constraints in the reverse process, and how these can be attained using a gradient-based projection method.*
>
> Finally, we obviously agree with your point about using the score function to increase tractability, but this does not change the overall objective from an optimization standpoint. Indeed again our method (Algorithm 1) directly employs the update step (line 6) used in [24] (with the variation of using a projected gradient version).
>
> We thank the reviewer for the opportunity to clarify these points. It is our intent to update the manuscript to better explain our derivation of Equation 3a; thank you for the suggestion.
>
> > Indeed, our implementation in Section D (Algorithm 2) ...
>
> We apologize if our response was unclear here. We reference Algorithm 2 because it dynamically adjusts the gradients to improve convergence; however, we do not explicitly attempt the learning rate schedules referenced in the question.
>
> ---
> Again, thank you for your willingness to engage with us during the discussion phase. We hope our responses clarify your points. Are there any additional concerns that we could address? We are ready to provide further insights as needed.
>
>
> ---
> [A] Welling, Max, and Yee W. Teh. “Bayesian learning via stochastic gradient Langevin dynamics.” Proceedings of the 28th international conference on machine learning (ICML-11). 2011.
>
> [B] Xu, Pan, et al. “Global convergence of Langevin dynamics based algorithms for nonconvex optimization.” Advances in Neural Information Processing Systems 31 (2018).
>
> [C] Raginsky, Maxim, Alexander Rakhlin, and Matus Telgarsky. "Non-convex learning via stochastic gradient langevin dynamics: a nonasymptotic analysis." Conference on Learning Theory. PMLR, 2017.

---

> > ### Comment · Reviewer_Jdpw · 2024-08-13
> > **Thanks again**
> >
> > Thanks to the references you provided, I now understand how the reverse diffusion process converges to its Gibbs distribution, which concentrates on the maximum likelihood solution. I have increased the soundness score from 1 to 2. I also agree that considering the reverse diffusion process as an optimization problem is a significant contribution of this paper. However, it is a pity that the introduction of this contribution is very vague, described as "akin to maximizing the density function." Although the author mentioned a plan to revise this, I hesitate to strongly accept the paper without seeing how the revision will be made, so I would like to keep my score.

---

> ### Author Response · Authors · 2024-08-13
>
> Thank you for your continued efforts to review our work. We are glad the additional clarifications have assisted in your understanding of our method.
>
> Do we understand that there are no more limitations pending and that as you mentioned our work is sound, novel, and significant? As you have stated that the only remaining hesitation for you to increase your score is that you would like to see our revisions, let us provide the changes we intend to make:
>
> - We will expand the beginning of Section 4 explaining our derivation of Equation 3a. We will begin by introducing the update step Equation 4 and explaining briefly explaining SGLD. This will be verbatim:
>
> >The application of the reverse diffusion process of score-based models is characterized by iteratively fitting the initial noisy samples $x_T$ to the learned approximation of $p(x_0)$.
> This optimization is formulated such that a variation of the traditional gradient descent algorithm, *Stochastic Gradient Langevin Dynamics* (SGLD), is used to iteratively transform a sample from the Gaussian distribution $q(x_T | x_0)$ to a sample from the learned distribution $q(x_1 | x_0)$. The update step is provided by:
> \begin{equation}
>     x_{t}^{i+1} = x_{t}^{i} + \gamma_t \nabla_{x_{t}^{i}} \log q(x_{t}^i|x_0) + \sqrt{2\gamma_t}\epsilon,
> \end{equation}
> where $\epsilon$ is standard normal and  $\gamma_t > 0$ is the step size. This step is repeated $M$ times for each $x_T$ to $x_0$. To prevent a deterministic behavior, an additional term is added to the gradient descent algorithm, $\sqrt{2\gamma_t}\epsilon$, drawing from *Langevin Dynamics* \cite{song2020score}.
>
> >SGLD can be viewed as an extension of traditional gradient descent algorithms, where the primary goal is to minimize a specified objective function. However, SGLD incorporates a stochastic component, introducing noise into this process. Formally, this procedure converges to a region characterized as an "almost-minimizer" of the objective function, with proximity to the minimizer bounded by $\frac{d^2}{(\sigma^{1/4}\lambda^*)}\log(1/\epsilon)$, where $\sigma^2$ represents the variance schedule, $\lambda^*$ denotes the uniform spectral gap of the Langevin diffusion, and $d$ is the dimensionality of the problem, as outlined in reference [C].
>
> >In this framework, the SGLD algorithm yields samples that are statistically concentrated around the global maximum of the underlying density function, as noted in [B].  Thus, the reverse diffusion process can effectively be approximated by minimizing the negative log-likelihood $-\log q(x_t|x_0)$, or equivalently as maximizing the density function $\log q(x_t|x_0)$ at each given noise level, the gradients of which are estimated by $s_\theta (x_t^i, t)$.
>
> >In traditional score-based models, at any point throughout the reverse process, $x_t$ is _unconstrained_.
> When these samples are required to satisfy some constraints, the objective remains unchanged, but the solution to this optimization must fall within a feasible region $C$,
> \begin{equation}
> \min_{x_{T}, \ldots, x_1} \sum_{t = T}^{1} -\log q(x_{t}|x_0)
> \end{equation}
> \begin{equation}
> \text{s.t.} \quad x_{T}, \ldots, x_0 \in C
> \end{equation}
>
> > Operationally, the negative log likelihood is minimized at each step of the reverse Markov chain, as the process transitions from $x_T$ to $x_0$. In this regard, and importantly, the objective of the PDM's sampling process is aligned with that of traditional score-based diffusion models.
>
> > To avoid low density data regions, the sample is optimized to conform to the previous distribution in the Markov chain before proceeding to the consecutive distribution, the transitions being  ensured by setting $x_{t-1}^{0} = x_{t}^{M}$, where as $x_t^M$ is the final iterate of the previous time step.
>
> - Following this, we will continue with Section 4.1.
>
> As we believe this is a fairly simple change, requiring rearranging some text and adding a few short paragraphs, we do not mind providing this. We hope that this summary of our intended revisions, provided at your request, addresses the remaining doubts. We also notice that adding this paragraph several to only strengthen our contribution, in addition to the strong empirical and theoretical support provided for what we believe is an important contribution for the adoption of diffusion models in scientific and engineering domains. As you have stated that this is the only remaining concern, we hope that you will consider advocating for strong acceptance.

---

### Author Response · Authors · 2024-08-04

Dear Reviewers and AC,

It has come to our attention that at this point in time some comments may be visible to the reviewers while others will remain hidden until the end of the author rebuttal phase (on Tuesday). This clarification was shared today by the NeurIPS PCs as this differs from the rebuttal process for many other venues. We apologize for any confusion this may have caused. The complete responses will be available when the rebuttals are made visible.

Thank you all for your understanding!

---

### Decision · Program_Chairs · 2024-09-25

**Decision:**

Accept (poster)

**Comment:**

The paper was an interesting application of ideas from diffusion and constrained optimization to have more controllable generative models that can satisfy some known physical principles/constraints.
The reviews overall are a bit borderline. Some of it stems from the paper subject itself which lies at an intersection of modern ML methods and applications to practical problems. The method being being more expensive than an unconstrained setup is likely irresolvable. The paper does report a fair variety of metrics to understand how well the method works. The method introduced is quite general purpose and should find wider applicability and open-sourcing the implementation would be valuable to the community.